# Overestimation in LLM Evaluation: A Controlled Large-Scale Study on Data Contamination's Impact on Machine Translation

**Muhammed Yusuf Kocyigit** [1]  **Eleftheria Briakou** [2]  **Daniel Deutsch** [2]  **Jiaming Luo** [2]  **Colin Cherry** [2]
**Markus Freitag** [2]

## Abstract

Data contamination—the accidental consumption of evaluation examples within the pre-training data—can undermine the validity of evaluation benchmarks. In this paper, we present a rigorous analysis of the effects of contamination on language models at 1B and 8B scales on the machine translation task. Starting from a carefully decontaminated train-test split, we systematically introduce contamination at various stages, scales, and data formats to isolate its effect and measure its impact on performance metrics. Our experiments reveal that contamination with both source and target substantially inflates BLEU scores, and this inflation is $2.5\times$ larger (up to 30 BLEU points) for 8B compared to 1B models. In contrast, source-only and target-only contamination generally produce smaller, less consistent over-estimations. Finally, we study how the temporal distribution and frequency of contaminated samples influence performance over-estimation across languages with varying degrees of data resources.

## 1. Introduction

Scaling laws have reshaped our understanding of the data requirements for training language models (LM), leading to a rapid expansion in data collection efforts. This expansion has inadvertently increased the probability of evaluation data contamination (Sainz et al., 2024): fragments or even entire test sets are accidentally consumed within the pre-training data, thereby invalidating the assumption that models are evaluated on unseen data.

Although several studies have acknowledged the contamination issue and shown that it can contribute to performance overestimations (Zhou et al., 2023; Jiang et al., 2024; Yang et al., 2023), we still lack a large-scale controlled analysis to better characterize the phenomenon. Concretely, prior efforts are commonly limited to smaller scales, both in terms of model and data (Jiang et al., 2024), treating contamination as a fine-tuning (Yang et al., 2023) or extended pre-training problem (Zhou et al., 2023), rather than addressing it directly in the larger-scale pre-training setting. Meanwhile, computational and logistical constraints have deterred large-scale, systematic experiments that would clarify how contamination interacts not only with model size, but also with training dynamics, and data composition.

In this paper, we present a controlled study to isolate and measure the impact of contamination during pre-training under diverse contamination conditions and two model scales (1 and 8 billion parameters). For this purpose, we take machine translation (MT) test suites as our case study, covering eight languages, thereby gaining insight into the interaction between contamination and linguistic resource availability.

Starting with a multilingual mixture of public pre-training corpora, we first decontaminate our train-test set splits and train a base LM on the training data (Figure 1). Then, we systematically introduce MT test examples into pre-training by varying their *mode* (presenting source and target in isolation or as full, prompted parallel texts), *temporal distribution* (controlling for when contamination happens), and *frequency* (controlling for the number of contamination copies). To efficiently explore various contamination conditions without the prohibitive cost of training from scratch, we adopted a branching strategy—resulting in more than 50% reduction in our training computational budget. Concretely, each experiment branched from a base model checkpoint and continued training on a modified data mixture, which replaces original samples with test contamination instances. Following our branching strategy, we studied a total of 42 contamination conditions for each model scale, by comparatively studying the performance of contaminated and uncontaminated models across contaminated and non-contaminated datasets. Our findings are summarized as follows:

[1]Boston University, USA, Work done while at Google. [2]Google. Correspondence to: Muhammed Yusuf Kocyigit <kocyigit@bu.edu>.

*Proceedings of the $42^{nd}$ International Conference on Machine Learning*, Vancouver, Canada. PMLR 267, 2025. Copyright 2025 by the author(s).

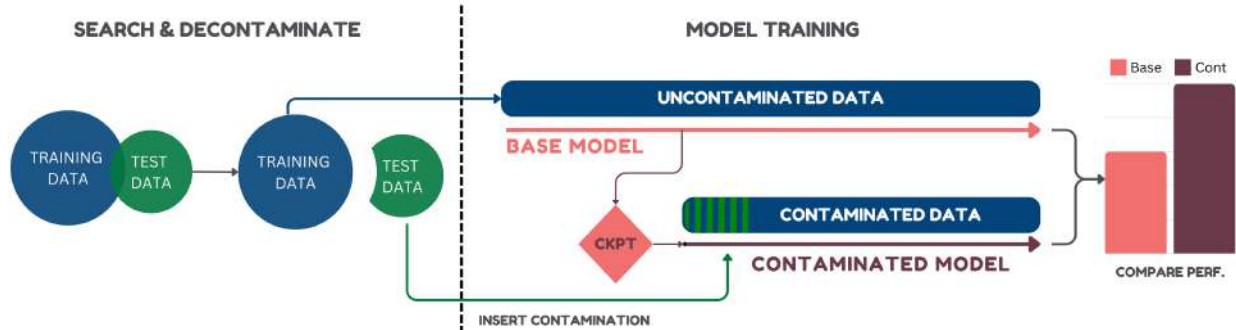

*Figure 1.* Large-scale contamination analysis setup: We decontaminate our train-test splits and train a baseline model. Then, insert test data into the pre-training data and train a contaminated model branching out from the baseline checkpoint. Finally, we compare the relative performance of the contaminated and the baseline model on contaminated and non-contaminated data.

- **Contaminating source-target MT pairs inflates performance on those test sets.** Contaminating examples with both source and target sides of the test data leads to substantial performance inflation—up to 30 BLEU points–for 8B-parameter models. On the other hand, partial contamination (source-only or target-only) generally yields smaller and less consistent inflations, especially when the improvements are compared to results on non-contaminated test sets (§5.1).

- **The temporal distribution of contamination matters.** Contaminating datasets at concentrated points causes large performance inflation at the time of contamination—up to 60 BLEU points—with their effects diminishing as training continues. On the other hand, uniformly introducing contamination throughout training has the most persistent impact (§5.2).

- **The impact of contamination increases with model scale.** Larger models exhibit increased sensitivity to even a single copy of contamination. Increasing the frequency of source-target contaminated examples yields higher performance overestimations, while source-only or target-only contamination is less affected by the frequency of contamination (§5.3).

- **Contamination requires sufficient language representation to have a measurable effect.** In the absence of language representation in the pre-training data, contamination showed no measurable impact on the languages and model scales studied (§5.5). Once a certain threshold of language representation is achieved, contamination has a larger impact on language-pairs with lower base model performance (§5.6).

## 2. Related Work

Prior work has studied contamination mainly by detecting it post-hoc or training models with contaminated data to understand its impact on model performance. The former involves examining token probabilities (Shi et al., 2024), analyzing model preferences for specific orderings (Oren et al., 2023) or analyzing model behavior with trained reference models to detect contamination or membership (Hisamoto et al., 2020). The latter assesses contamination's impact by intentionally contaminating test sets into LLMs' pre-training mixtures (Jiang et al., 2024; Yang et al., 2023; Magar & Schwartz, 2022; Zhou et al., 2023). Finally there is is also work that suggest model developers provide the community with search tools for downstream inspection of membership (Marone & Durme, 2023).

We present a comparative summary of our method and previous work in Table 1. Several studies have explored the impact of data contamination by extended pre-training or fine-tuning on contaminated data (Table 1, Training Process), leaving the generalization of their results to contamination in pre-training an open question. Zhou et al. (2023) set up their experiments as extended pre-training, which could overstate contamination's impact, as models are more susceptible to performance inflation when contamination is introduced later in training (Magar & Schwartz, 2022). On the other hand, studying contamination during fine-tuning, as in Yang et al. (2023), involves substantial training changes—from learning rate to batch size choices— further complicating its generalizability to pre-training.

Previous works analyze models and pre-training corpora of limited sizes (Table 1, Model Size and Data Size), primarily restricted by resource constraints. For example, Jiang et al. (2024) use models ranging from 124M to 774M parameters, while Magar & Schwartz (2022) use models with 110M to 345M parameters. Our work contributes an analysis at larger scales (1B and 8B parameters and 325B tokens) to shed more light on how contamination behaves at scale.

Moreover, methods measuring the impact of contamination or detecting it via model activations often assume a well-understood pre-training mixture and that any unintended

| Method | Training Process | Data Control | Model Size | Data Size | Multilingual |
|--------|------------------|:------------:|:----------:|:---------:|:------------:|
| Zhou et al. (2023) | Extended pre-training | ✗ | 1.3B, 7B | NA | ✗ |
| Yang et al. (2023) | Fine tuning | ✗ | 7B, 13B | NA | ✗ |
| Magar & Schwartz (2022) | Pre-training | ✗ | 110M, 345M | 600M | ✗ |
| Jiang et al. (2024) | Pre-training | ✗ | 124M, 774M | 20B, 60B | ✗ |
| Ours | Pre-training | ✓ | 1B, 8B | 325B | ✓ |

*Table 1.* A comparison of related work to our experimental setup. "Data Control" reflects whether the word checks for existing training-test contamination. Unlike prior work, we study contamination during pre-training in a controlled, large-scale, multilingual setting.

data is absent from the corpus (Table 1, Data Control). Consequently, prior work usually omits a rigorous decontamination step to establish a clean base model performance (Jiang et al., 2024; Magar & Schwartz, 2022; Zhou et al., 2023). However, existing work indicates that contamination frequently occurs in public pre-training corpora, highlighting the importance of careful data control (Singh et al., 2024).

Finally, past work focuses on reasoning, math, question answering, summarization, and coding tasks to assess contamination effects. To our knowledge, our study is the first to address this issue on machine translation, allowing us to examine contamination across a diverse set of language resource availability (Table 1, Multilingual).

## 3. Large-Scale Contamination Analysis

Our experimental pipeline comprises four steps (Figure 1). We start by searching the pre-training mixture with an n-gram search algorithm to detect existing overlap with our evaluation datasets (§3.1). Then, we decontaminate our test set, and train a baseline model on the training split (§3.2). This baseline is used as a reference, uncontaminated model in our experiments. As a next step, we systematically contaminate MT evaluation sets within the baseline's pre-training mixture by defining a wide range of contamination conditions (§3.3). Finally, to efficiently manage the large-scale nature of our analysis, we developed a checkpoint-branching approach, resulting in a jungle of contaminated checkpoints, which are compared against the baseline to isolate the effect of contamination (§3.4).

### 3.1. Search and Decontaminate Test Sets

We implement an 8-gram search to find matches between the test sets and the pre-training data. We do not normalize the text and work with sub-word tokens instead of white space-split text. We search for source and target contamination separately. An example is labelled as contaminated if the longest matching sub-sequence (Singh et al., 2024) matches more than 70% (Chowdhery et al., 2022) of their source or target tokens. By running this algorithm against our pre-training mixture, we found around 10% of test examples being already contaminated. We removed those examples

from our test sets to ensure a clean setup. Detailed statistics are found in Appendix G.

### 3.2. Uncontaminated Baseline Model

Our 1B and 8B models are decoder-only transformer models (Vaswani, 2017) trained with a casual language modelling objective. We use a sentence piece tokenizer (Kudo, 2018) with a vocabulary size of 256K. We use a $4,096$ token context window and a batch size of 512 for 155K steps, bringing our training budget to 325B tokens. The 8B and the 1B models are trained using the same hyper-parameter settings and data.[1] During training, we track loss on the validation set—a small random sample from the training mixture. We use the ADAM (Kingma & Ba, 2014) optimizer with cosine learning rate decay with a warmup phase.

### 3.3. Contamination Conditions

We experimented with various ways of contaminating MT samples along three dimensions (Table 2). First, we define different *modes* of contamination—one where a sample is presented as a full, prompted[2] source-target instance and two partial contamination cases where the model is only exposed to either the source or the target. For source-target contamination, we introduce two additional variants by injecting each side as independent, unformatted samples either within or across different batches. Second, we vary the temporal distribution of contaminated samples by injecting them at different pre-training points or uniformly distributing them throughout. Third, we vary the number of times each contaminated sample is presented to the model.

Contaminated samples are mixed with existing training data, ensuring they are not seen in isolation. Contamination for each setting is introduced within a pre-determined window of training steps to enable fair comparisons.

---

[1] The selected training budget is likely too large for the 1B model; however, keeping the training data the same for both the 8B and 1B makes isolating the impact of scale possible.

[2] Prompted examples are formatted by prepending the language name in English as shown below:
"German: Diego Cocca is gut.
English: Diego Cocca is good."

| Parameter | Values |
|---|---|
| Mode | Source, Target, Full |
| Temporal Distribution | Early (30% of training), Middle (60% of training), Late (90% of training), Uniform (randomly distributed between 30–90% of training) |
| Frequency | 1 , 10 or 100 copies |

*Table 2.* Contamination conditions studied in this work.

### 3.4. Checkpoint-branching

To avoid training multiple LLMs from scratch, each contamination setting branches out from the baseline checkpoint and continues pre-training by inheriting all its training hyperparameters (see Figure 1). Here each new branch is a copy of the baseline model checkpoint and continues training on a modified copy of the original training data where the only change is the added contamination data. So essentially each experiment is a variant that inherits the past from the same baseline model. This gives us two main benefits: *efficiency*—by avoiding training from scratch for each contamination setting, we reduced the total training budget by 53.6%, and **reduced variance** between contaminated and baseline runs—allowing us for better isolating the impact of contamination by increasing the overlap in the compared models' training and initialization.

## 4. Experimental Setup

**Pre-training Data** Our pre-training data are drawn from multiple *public* resources. Monolingual texts are sourced from Dolma (Soldaini et al., 2024) for English and Madlad (Kudugunta et al., 2024) for covering non-English languages. In addition to monolingual resources, we add parallel data into our pre-training mixture, which has shown to be critical in enabling translation capabilities at the scales we study (Briakou et al., 2023; Chowdhery et al., 2022; Alves et al., 2024). Our parallel data are sourced from the WMT'23 translation task (Kocmi et al., 2023). The total size of these datasets is about 2T tokens, which we have downsampled for our purposes. The total pre-training mixture consists of 325 billion tokens based on our multilingual sentence-piece tokenizer. The different sources are mixed based on the ratios of 60% for Dolma, 35% for Madlad, and 5% for parallel texts. The exact token counts and Madlad languages can be seen in Table 3. We randomly sampled from Dolma and Madlad sources without altering their domain or language distributions. All parallel texts from WMT are used and up-sampled to fit our 5% parallel data budget. When we insert contamination into the data, we try to keep the ratio

of parallel data constant by only randomly replacing less than 5% of the examples in a batch with contamination.

| Data | Number of Tokens |
|---|---|
| Dolma | 195, 035, 136, 000 |
| Madlad (*total*) | 113, 770, 495, 994 |
| ↪ RU | 56, 044, 201, 738 |
| ↪ DE | 30, 535, 489, 020 |
| ↪ JA | 8, 909, 192, 431 |
| ↪ ZH | 5, 722, 035, 885 |
| ↪ CS | 5, 671, 961, 832 |
| ↪ UK | 3, 634, 722, 726 |
| ↪ AR | 1, 810, 191, 105 |
| ↪ HE | 1, 442, 701, 257 |
| Parallel data | 16, 194, 368, 006 |
| **Total** | 325, 000, 000, 000 |

*Table 3.* Pre-train mixture statistics (measured as token counts).

**Evaluation Data** We evaluate our models on both *non-contaminated* and *contaminated datasets*. All tests are run with the same prompt format. The contaminated datasets are sourced from WMT 2023, covering 10 language pairs: EN-DE, EN-RU, EN-CS, EN-UK, EN-HE, DE-EN, RU-EN, UK-EN, HE-EN, and CS-UK. As our non-contaminated tests we use WMT 2024 (Kocmi et al., 2024) for the five overlapping language-pairs with our contaminated sets (EN-DE, EN-RU, EN-UK, EN-CS, and CS-UK). Moreover, we include three low-resource languages, Achinese in Arabic script,[3] Wolof and Yoruba from FLORES (Goyal et al., 2022), which we assume are fairly zero resources within our pre-training data. During our decontamination stage, 681 examples from WMT'23 are removed, leaving us with 9, 491 examples and 900 examples from FLORES are removed, leaving us with 2, 136 examples.

**Evaluation Metrics** We evaluate translation quality using both string-based metrics—BLEU (the main paper) (Papineni et al., 2002) and top-performing learned neural metrics—MetricX (Appendix B) (Juraska et al., 2023) as assessed by the WMT metric shared task (Freitag et al., 2023).

## 5. Results

**Main Findings** We present an overview of our results in Figure 2, which shows the absolute BLEU score differences between contamination models of different flavors, each compared against the baseline, uncontaminated model on the WMT'23 contaminated datasets. Several clear trends emerge. First, contaminating both source and target in a prompted format (*Full*) into the training data—regardless of

---

[3] We chose Achinese in Arabic script to check if the script made any difference on the impact of contamination.

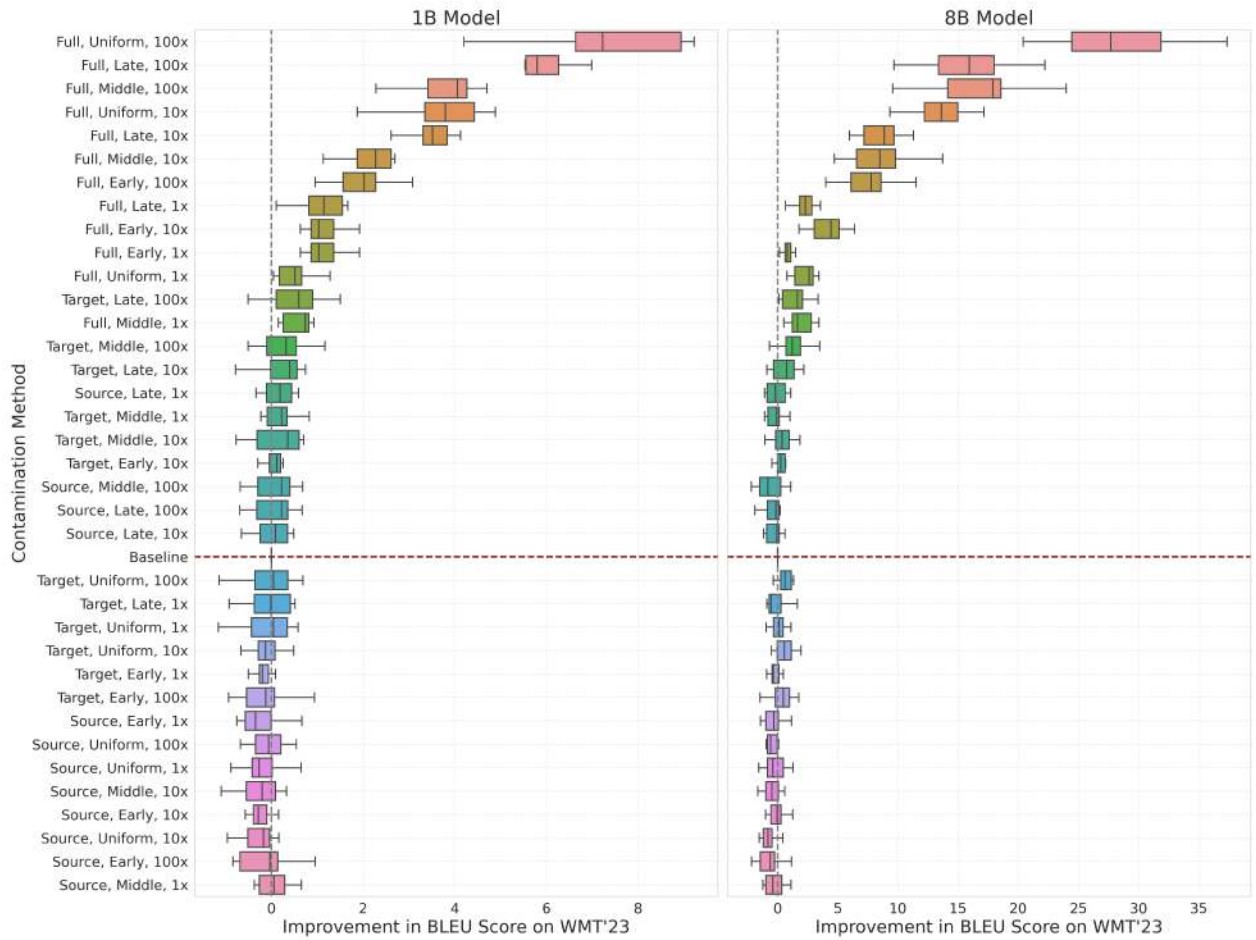

*Figure 2.* Box plot of BLEU differences of contaminated vs. uncontaminated models across WMT'23 language-pairs, for 1B (left) and 8B (rights) model sizes. Contaminating paired source-target instances (full) consistently inflates translation performance across languages, with larger effects on the 8B model. Source-only and target-only contamination does not inflate performance consistently.

when contamination happens during training or how many times the contaminated instances are seen by the model—consistently inflates model performance. Second, this inflation becomes more pronounced as model size increases. For the 1B model, maximum performance over-estimates hover around 9 BLEU points, while for the 8B model, they can reach up to 30 BLEU points. In fact, under *Full* contamination, the 8B model's performance inflation is, on average, $2.5\times$ as large as that of the 1B model.

Figure 2 also shows that source-only or target-only contamination does not universally boost BLEU scores across the languages studied. Although some settings yield positive mean and median improvements, these gains are neither consistent nor on the same scale as those observed with *Full* contamination. In other words, contamination involving only one side of the parallel data appears to be less critical from an evaluation standpoint.

## 5.1. Are performance over-estimations actually over-estimations?

The contaminated MT test examples are arguably a high-quality source of parallel texts. Therefore, it is reasonable to ask: to what extent does the increased performance come from contamination inflation rather than genuine improvements in the model's translation capabilities? If the latter was true, we would expect the performance improvements to generalize to other test sets of the same task that are not contaminated. To account for this, we evaluate our models on the WMT'24 non-contaminated test sets.

Figure 3 shows how the improvement differences are distributed across the five language pairs shared between WMT'23 and WMT'24, for each contamination setting. This comparison helps contextualise the contaminated improvements we discussed in Section 5. First, we notice that models exposed to source-only or target-only contamination show negligible improvement differences across the con-

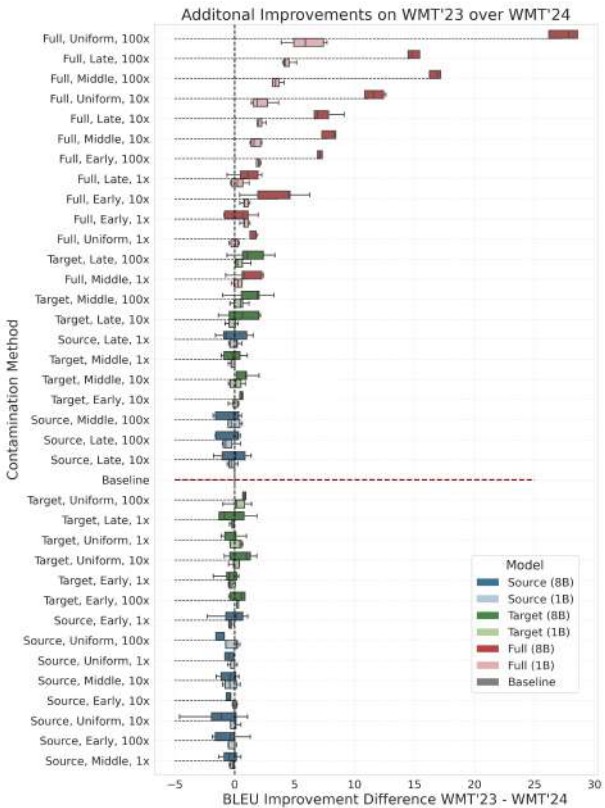

*Figure 3.* Box plot of BLEU improvement differences of contaminated vs. uncontaminated models between WMT'23 - WMT'24. Contaminating source-targe examples yields higher performance "improvements" on contaminated vs. non-contaminated datasets.

taminated and non-contaminated test sets. This indicates that partial contamination does not fundamentally distort performance estimates, at least at the scales we studied. In contrast, for models exposed to *Full* contamination, we notice that the average improvements on the contaminated datasets are up to 26 BLEU points larger than those observed in the non-contaminated sets. Finally, increasing the frequency of contaminated examples consistently increases the performance gap between the two contrasted datasets.

### 5.2. How does the temporal distribution of contamination impact performance inflation?

As discussed in §2, previous works studied contamination in extended pre-training or fine-tuning settings. However, as shown in Figure 4, the temporal distribution of contamination, i.e., the time-step(s) at which a pre-training model is exposed to contamination, impacts the inflation we observe at the end of pre-training. Concretely, we notice that the earlier the contamination is introduced, the larger the immediate spike in performance. As training continues, those earlier momentary spikes are rapidly wearing off, with their

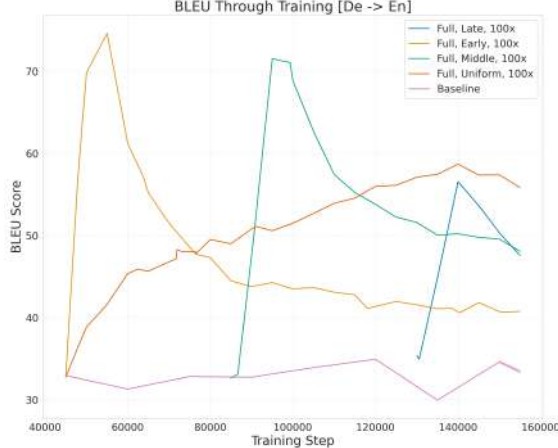

*Figure 4.* BLEU score throughout training for German to English in WMT'23 for the 8B model, *Full* contamination and 100 Copies. Earlier contamination causes larger performance peaks, while later contamination causes lower spikes but higher eventual performance gaps. Uniform contamination tends to yield the highest final performance gains and no sharp peaks.

ultimate effect being way smaller than their initial intensity, i.e., a spike of $\sim 70$ BLEU points is mapped to $\sim 40$ points after $\sim 100$K steps. As a result, observing contaminated data later during training has a bigger footprint than early exposure, which questions whether studying contamination within extended pre-training or fine-tuning settings exaggerates its measurable impact in real pre-training scenarios.[4] A potential explanation for this behaviour is that the learning rate is set to a higher value in earlier stages since it is decayed during training—a trend also noticed in Magar & Schwartz (2022).

Furthermore, we observe that uniform contamination— contaminated examples being uniformly spread out during pre-training instead of being presented within concentrated time steps—results in larger performance inflation across all temporal settings, even the one where contamination is exposed late at pre-training. This finding has important implications, given that uniform contamination reflects a more realistic contamination scenario in the wild, and randomizing the pre-training data order is common practice.

### 5.3. How does the frequency of contamination impact performance inflation?

Figure 5 presents how the performance of the contaminated models improves on the contaminated datasets as the frequency of contamination increases. As seen, for particular language pairs (dashed lines), performance inflation follows a Λ-Shaped curve, in line with what is observed in prior

---

[4]We also show that this trend is general across other language pairs, number of copies and model size in Appendix C.

works (Jiang et al., 2024). Although some languages follow this trend, most of them do not. On average, for *Full* contamination, the inflation from contamination increases with the number of copies. On the other hand, partial contamination exhibits a different trend as the performance does not improve with increased number of contamination copies.

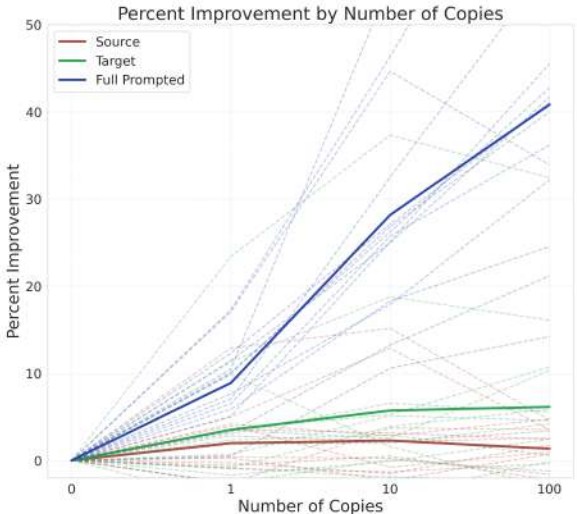

*Figure 5.* Percent improvement for different contamination methods for increasing number of copies for the 8B model. The dotted lines are the percentage improvements per language pair in WMT'23. The solid lines are the mean improvement per method. Performance inflation increases with more copies of *Full* contamination, while additional copies of source- or target-only contamination do not significantly alter the overall impact.

### 5.4. How is contamination format impacting performance inflation?

Contaminated data can be presented within pre-training mixtures in different formats depending on a variety of reasons, starting from how they naturally occur on the web to how curated data are pre-processed before consumed by LLMs. When it comes to contamination of source-target MT examples, we have so far explored the special case where the contaminated dataset is consumed as *Full*, formatted examples in the same prompt format used at test time. However, test data on the Internet is not necessarily stored in that same format, while source and target texts can be maintained in separate files, which means that source-target pairs can be contaminated as unpaired examples. To simulate such scenarios, we create two variations of source-target contamination: starting from a given source-target text; we insert each side as a separate, unpaired example into the same batch (named *Source and Target, Batched*) or in different batches (named *Source and Target, Split*).

Figure 6 compares these two settings with the prompted format (*Source and Target, Prompted also named Full*) and

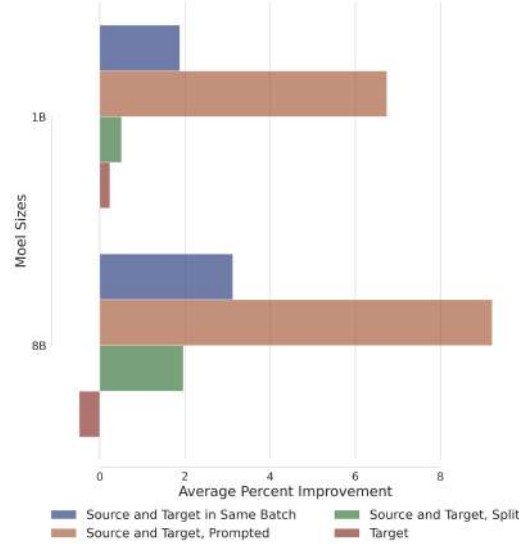

*Figure 6.* Average percentage performance overestimation for different ways of contaminating the target text (late contamination with 1 copy). Contaminating both the source and target as separate examples, whether within the same batch or at different positions during training, can cause greater performance inflation than contaminating only the target. Contaminating prompted source-target examples leads to the highest performance inflation.

the case where we only contaminate the target text which we add as an additional reference point. Comparing with these baselines, we see that both the *Split* and *Batched* settings perform better than target-only contamination however, the prompted format still results in higher inflation. Comparing *Split* and *Batched*, we see that consuming the unpaired texts within the same batch causes larger performance over estimations compared to spreading them across different batches, even though the examples are not presented as paired, prompted translation texts. We also observe that the additional performance inflation caused by *Batched* and especially *Split* contamination compared to target-only contamination is larger for the 8B model than the 1B model.

### 5.5. How does contamination impact performance of near zero-resourced languages?

To understand the impact of contamination for languages with no intentional language representation during pre-training, we contaminate MT examples from three languages—Achenese, Wolof and Yoruba—sampled from FLORES. We note that we intentionally do not include any monolingual or parallel data for those languages in our pre-training mixtures.

For these language pairs, we observed no performance inflation from contamination (Table 4). Even right at the contamination, the BLEU score increased only by 1 to, at

| Contam. | Copies | Ace-En | Yo-En | Wo-En |
|---------|--------|--------|-------|-------|
| ✗ | NA | 0.095 | 1.445 | 1.521 |
| ✓ | 1 | 0.328 | 1.204 | 1.417 |
| ✓ | 10 | 0.558 | 1.361 | 1.633 |
| ✓ | 100 | 0.722 | 1.345 | 1.776 |

*Table 4.* BLEU scores for non-contaminated and contaminated models on zero-resourced languages for 8B models. Contaminating test sets for languages with no representation during pre-training does not result in any performance inflation.

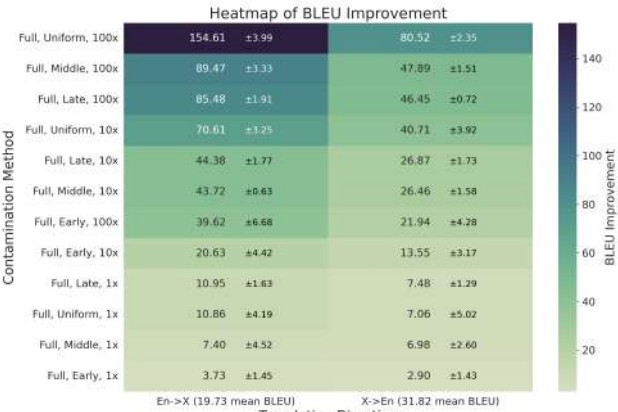

*Figure 7.* Average percentage BLEU score improvement for two performance groups, En→X and X→En. Performance improvements for the En→X are higher compared to the X→En direction.

most, 3 points for both the 8B and 1B models, as seen in Appendix D. This suggests that gains from contamination require the model having some of representation in that language, at least at the scales investigated in this work.

### 5.6. How does contamination impact performance into versus out of English directions?

Figure 7 illustrates the percentage BLEU score improvements grouped by out and into English translation directions (En→X and X→En, respectively). As shown, contamination has a more significant impact on the En→X translation direction compared to X→En, for all *Full* contamination settings. While we present percent improvements, the absolute improvements are also larger for the En→X direction for almost all cases. Considering the results from Section 5.5, one could naturally expect the model to benefit more in the X→En since it has better performance and hence better representations in English. Contrarily, we observe that the model can benefit more from contamination in En→X, where the original performance is lower. These results, together with our findings from Section 5.5, show that while a certain level of representation is necessary to benefit from contamination, the performance gains do not continue to constantly increase as model capabilities improve, highlighting the complex relationship between base model capabilities and the effects of contamination. This finding highlights the complex relationship between contamination and the availability of language resources and demonstrates how previously observed model behaviors under well resourced settings (English) might fall short when moving to multilingual settings.

## 6. Limitations and Discussion

Drawing definitive conclusions on whether contamination matters is very challenging due to sources of variance that are challenging to control for. Initialization seed, data ordering, among others, introduce variance into the analysis. In an ideal world, one would run each contamination experiment for multiple random seeds of the model and data orders. However, resource constraints deem this setup impractical.

Despite those challenges, our experimental setup takes steps to control for variance as much as possible. For instance, we fixed the order of the data and the model initialization. While these deterministic systems helped minimize random variations between runs and allowed us to isolate the impact of contamination, they also imposed certain limitations on our findings. Specifically, our results are based on a single canonical ordering of the training dataset and a single initialization of the two models used in the experiments.

Another source of variation we observed was between different checkpoints (stopping points). This pattern is further elaborated in the Appendix D, where the model's performance can vary for different language pairs between different training steps. This makes interpreting individual data points for language pairs at the end of training more challenging. To tackle this problem, we focused on analyzing meaningful aggregates and general trends rather than changes and variations for individual language pairs.

One challenge this introduced is that changes in BLEU score across different scales do not correspond to comparable differences in quality. This aggregated BLEU scores across different value ranges can be tricky and must be done carefully. To tackle this problem, we either use the same language pairs in all aggregates that are compared at a single experiment or when we split language pairs, we group language pairs that are on similar BLEU scales together. We also compare average BLEU changes when comparing WMT'23 and WMT'24 performances but acknowledge the caveat explained here.

Additionally all our experiments are done using decoder-only transformer models. We chose this architecture due to its prevalence however our findings may not generalize to other transformer types or architectures such as LSTMs. Fi-

nally, our experiments on models with up to 8B parameters indicate that the impact of contamination grows with model size, although this trend is not guaranteed to hold for larger models.

## 7. Conclusion

In this work, we study the impact of contamination on large language model pre-training, with a focus on the task of machine translation. Our work employs a checkpoint-branching strategy that allows us to efficiently scale up our study to 46 contamination conditions across two-model scales, 1 and 8 billion parameters, and 13 language-pairs. The key experimental results include that (1) contaminating both the source and target text leads to substantial performance inflation; (2) when the contamination is observed during training influences the size of its impact, with uniform contamination having the biggest effect; (3) larger models benefit more from contamination, and (4) contamination requires sufficient language representation to have a measurable effect. This work sheds light on the nuanced ways in which data contamination affects model performance, and underscores the need for more reliable evaluation practices in large language model development.

## Impact Statement

We recognize that the experiments in this paper required significant computational resources, which came with an environmental cost due to high energy consumption. We do try to tackle this by implementing a checkpoint branching approach however the experiments still require a large amount of compute. This was necessary to ensure robust and meaningful results, we hope that our findings will help reduce the need for repeated experiments in the future. By making our results available, we aim to contribute to the field in setting up better evaluation practices and support researchers in building on our work more efficiently, ultimately lowering the environmental footprint of similar studies in the long term.

This paper contributes to advancing Machine Learning. While our work may have also have broader societal implications, we do not believe any specific concerns need to be highlighted here.

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

# A. Appendix: Method of Contamination and Examples

In Table 5 we present the different method of contamination we insert into the pre-training data and how they are formatted. We use an example from German to English task. The first three methods are used for the main experiments while method 4 and 5 are used in the subsection 5.4

| Method | Definition | Example |
|---|---|---|
| Full Example | Insert the source and target text for all examples as separate formatted examples. Each example is presented exactly as they would appear during testing. | German: Diego Cocca wird neuer National-trainer von Mexiko
English: Diego Cocca will become the new national team trainer for Mexico |
| Just Source | Insert just the source text without any formatting as separate examples The source texts are presented as standalone examples int the pretraining data. | Diego Cocca wird neuer Nationaltrainer von Mexiko |
| Just Target | Insert just the target text without any formatting as separate examples The target texts are presented as standalone examples int the pre-training data. | Diego Cocca will become the new national team trainer for Mexico |
| Source with Target Split | Source and Target are prompted just like Just Source and Just Target, but the examples the source and target are inserted in separate locations. | Diego Cocca wird neuer Nationaltrainer von Mexiko
⋮
Diego Cocca will become the new national team trainer for Mexico |
| Source with Target Batched | Source and Target are prompted just like Just Source and Just Target, The source and target for the same example are consumed as separete inputs in the same batch. | Diego Cocca wird neuer Nationaltrainer von Mexiko
Diego Cocca will become the new national team trainer for Mexico |

*Table 5.* The types of formatting done for the contamination inserted into the pre-training corpora.

# B. Appendix: MetricX Counterparts of Main Figures

We present a MetricX counterpart of the main figures in the results section in below. In Figure 8 we present the MetricX improvements per contamination method. Here our observations are similar to that of the main section. We observe that the improvements for the larger model are much higher. We also see that *Full* contamination improves performance across the board while source-only and target-only contamination don't show consistent over-performance. The difference in performance gain with the larger model is smaller when observed with MetricX this is likely due to the limited scale MetricX is measured.

We also present how the MetricX values change over the cours of training for one language pair in Figure 9 and how the number of copies impacts performance over-estimations for MetricX 10.

Finally we also present our analysis in §5.6 in MetricX and show that our general findings hold true for MetricX as well as BLEU

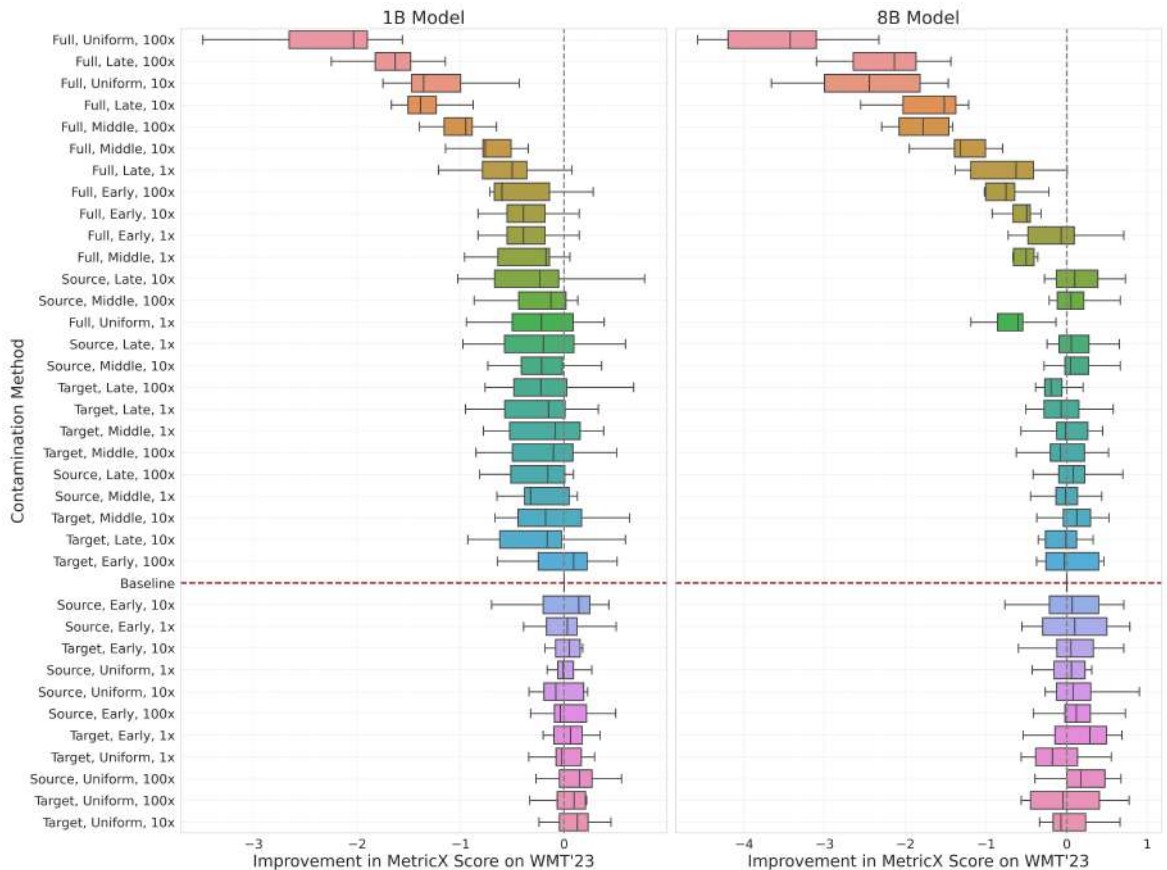

*Figure 8.* Box plot of absolute MetricX (lower is better) improvements for all 10 WMT'23 Language Pairs for 1B and 8B Model. Notice the scales of the X-axis is different for different model sizes. The methods on the Y-axis are sorted based on the mean improvement for the 1B model. Methods above the red line demonstrate positive mean and median (central line in the box plot) improvements for the language pairs that are considered. Since lower is better for MetricX, a larger negative score in this plot means a bigger improvement.

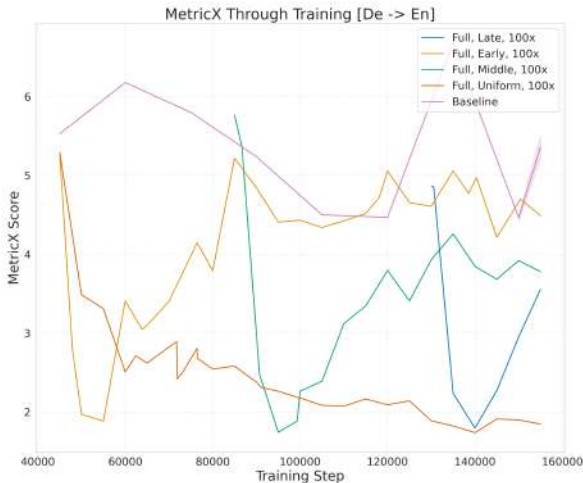

*Figure 9.* MetricX (lower is better) score throughout training for German to English in WMT'23 for the 8B model, *Full* contamination and 100 Copies.

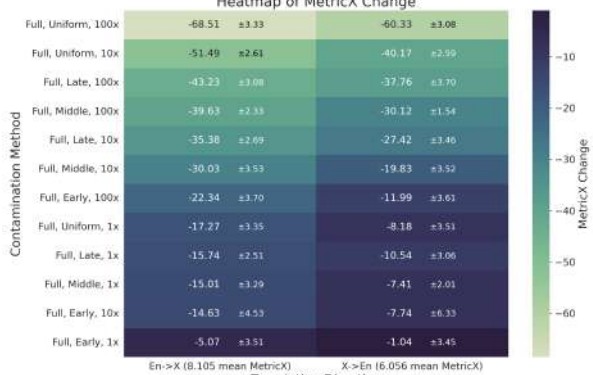

*Figure 11.* Average percentage MetricX score improvement for two performance groups.

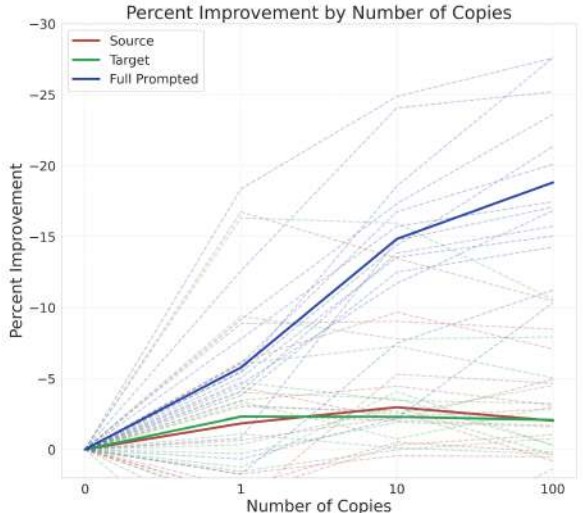

*Figure 10.* Percent improvement for different contamination methods for increasing number of copies. Dotted lines are the percentage improvements per language pair in WMT'23. The solid lines are the mean improvement per method.

## C. Appendix: Position of Contamination

We also share some analysis on the position of contamination. We summarize the impact of contamination for *Full* contamination in early, middle, late and uniform contamination for 1,10 and 100 copies in Figure 12. This representation also supports the findings of Section 5.2.

The same data can also be analyzed from the position of WMT'24. In this case this isn't contamination but high-quality task-related data. So Figure 13 can be read from the position of generalization from high quality data. Though these findings should be interpreted cautiously, we see some interesting patterns.

First inserting more copies has diminishing impact especially after 10 copies. When including high quality data later in the stage both improves generalization performance at the time of seeing the data as well as at the end of training. This could be a function of the model being under/over trained and over trained models seem to be benefiting more from high quality data. This holds both if stop training after seeing the high quality data or if we keep on training on other data as well.

This pattern can also be observed when we look at the performance over training plots in Appendix D and Appendix E. We see that the peaks in WMT'23 are highest in early contamination while the peaks in WMT'24 are highest with late contamination. This suggests that an overtrained model generalizes better from high quality data compared to a comparatively undertrained model even if the learning rate is smaller.

Finally uniformly spreading high quality data seems to give the highest generalization improvements. However this could similarly be a result of gradient clipping applied when the test data is introduced in a single location.

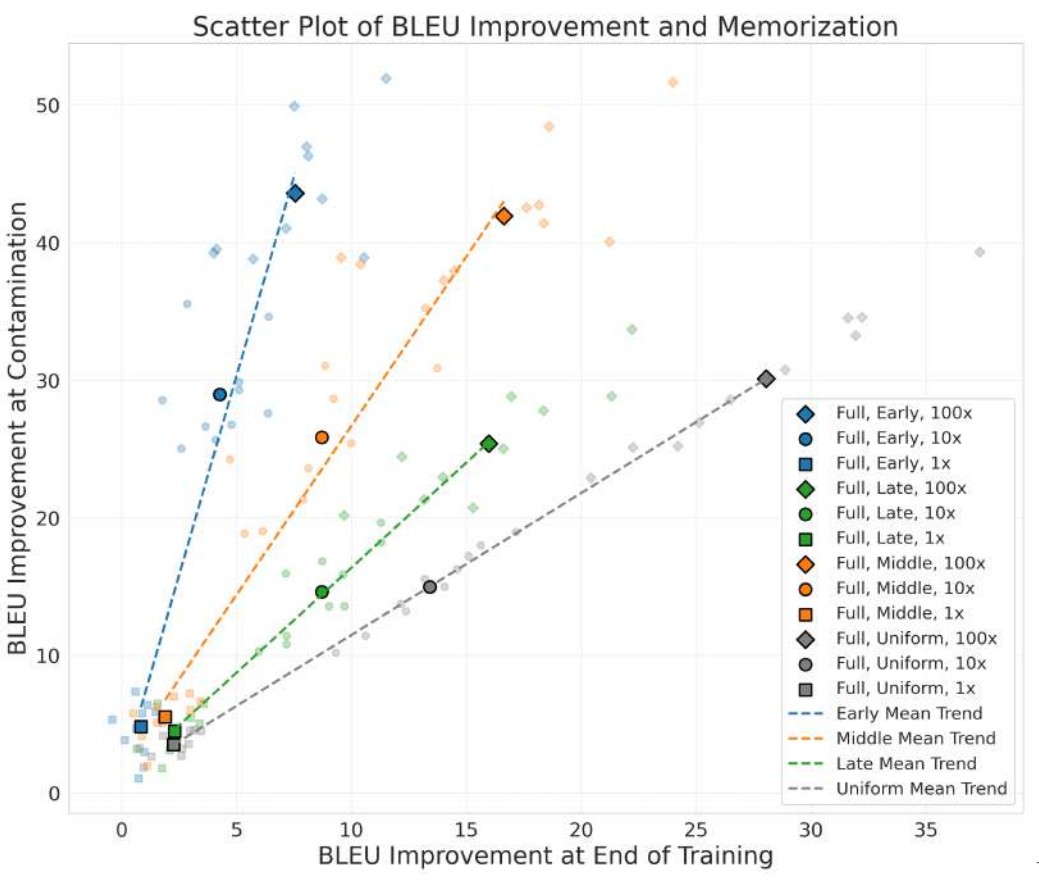

]

*Figure 12.* The performance improvement right after the contamination and performance improvement at the end of training for WMT23 language pairs. We show that our observations in Section 5.2 hold true in general for all WMT'23 langauge pairs and number of copies.

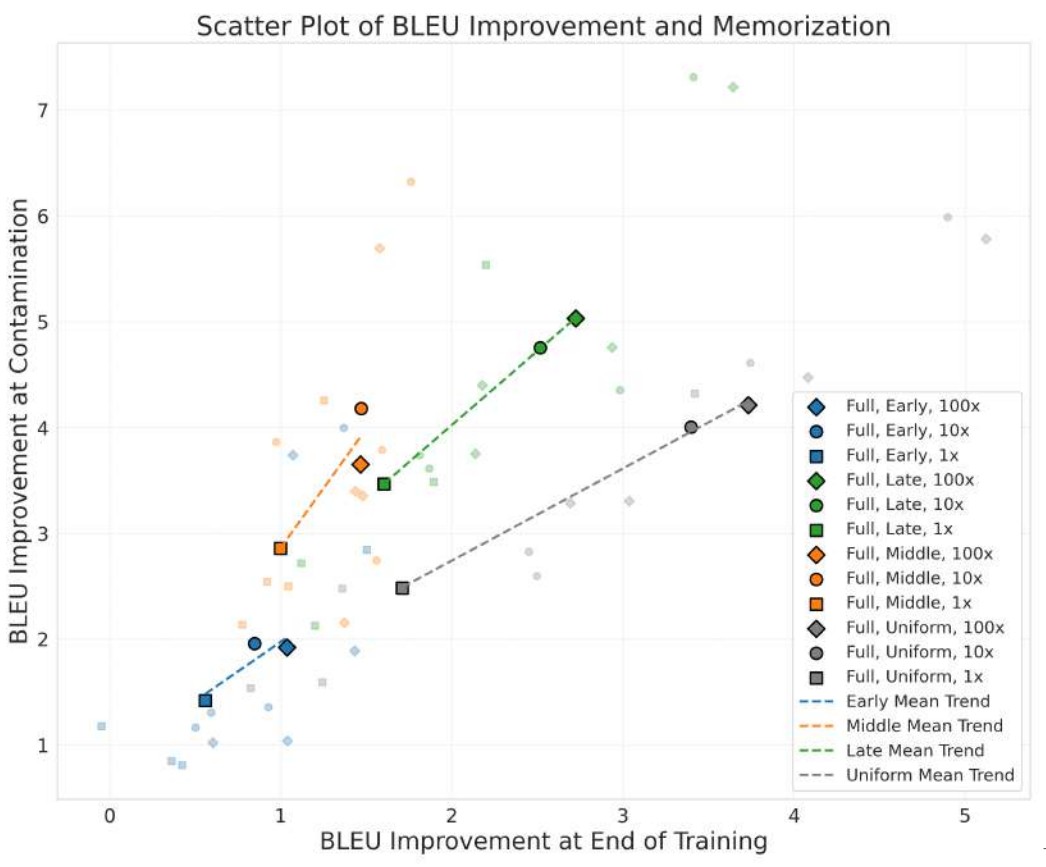

]

*Figure 13.* The performance improvement right after the contamination and performance improvement at the end of training for WMT24 language pairs. In this context we can read this plot in the context of adding high-quality data into the pre-training mixture to see how it improves performance. We observe that inserting it early causes neither large instantaneous performance boosts nor does it cause it large performance differences in the eventual performance. For late contamination we see that both the peaks are higher and the eventual performance boost is also more finally with uniform contamination the eventual performance gain is the highest. We also observe that for figure 4 the difference between 10 and 100 copies was much larger while for here we see that there isn't much difference between 10 and 100 copies which is natural.

# D. Appendix: BLEU Scores Through training plots

Below we present the BLEU score across training for all the language pairs that are present in WMT'23 and the FLORES langauge paris that we have included in the contamination.

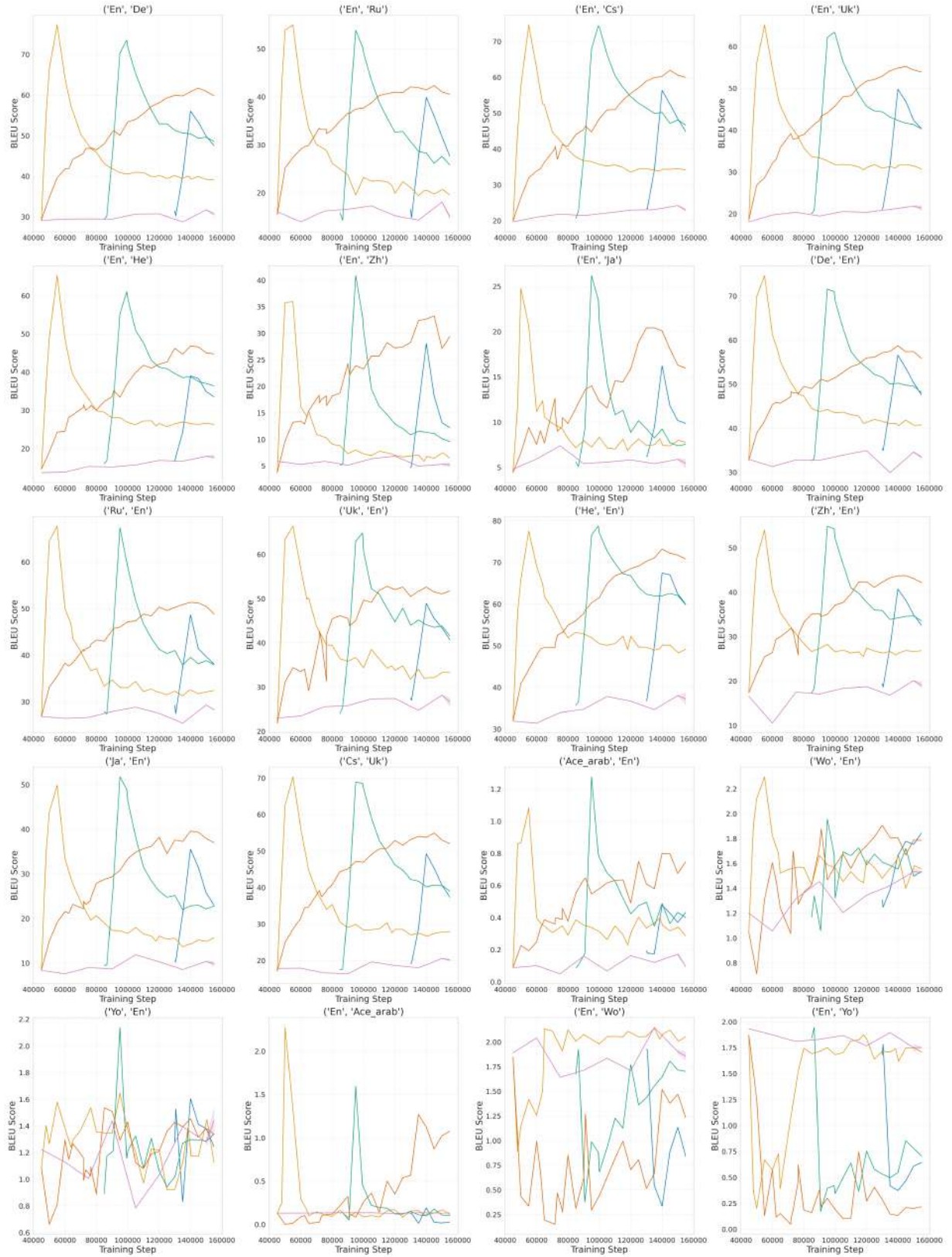

*Figure 14.* BLEU scores through training for 100 Copies of *Full* contamination

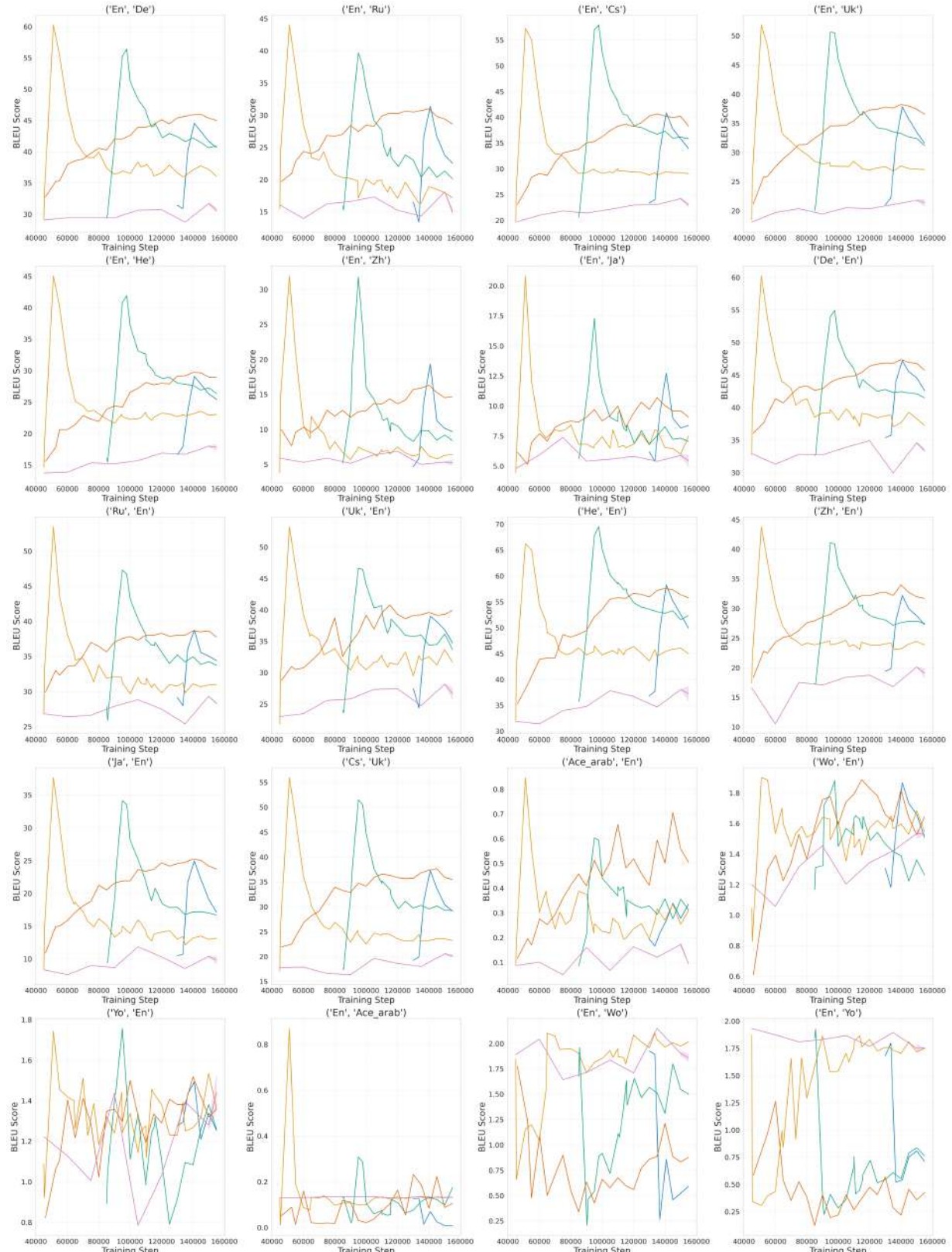

*Figure 15.* BLEU scores through training for 10 Copies of *Full* contamination

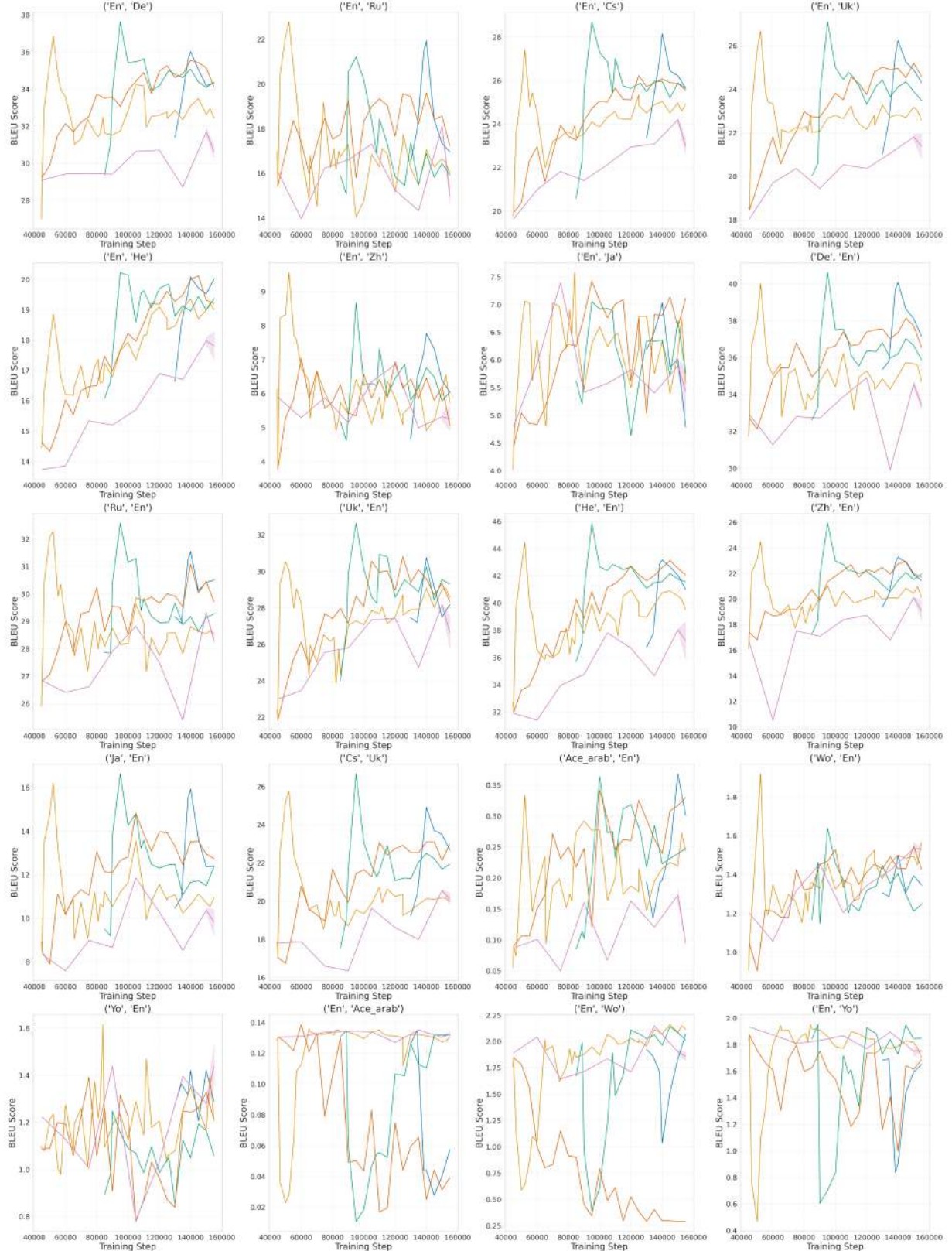

*Figure 16.* BLEU scores through training for 1 Copies of *Full* contamination

# E. Appendix: Results of WMT24 Experiments

| | 1B Model | | | | 8B Model | | | |
|---|---|---|---|---|---|---|---|---|
| | Baseline | Source | Target | Full | Baseline | Source | Target | Full |
| **EN → X** | | | | | | | | |
| De | 17.39 | **17.77** | **17.66** | **18.43** | 24.01 | **24.02** | **24.39** | **25.13** |
| Ru | 10.57 | **10.86** | **11.12** | **11.69** | 13.23 | **13.94** | **13.90** | **15.43** |
| Uk | 11.40 | **11.42** | 11.28 | **12.31** | 18.93 | 18.86 | **19.68** | **20.82** |
| Ja | 12.38 | **13.92** | **14.22** | **15.50** | 17.60 | **19.30** | **20.71** | **19.89** |
| Zh | 5.74 | **6.18** | **6.36** | **6.46** | 5.53 | **5.65** | **5.83** | **6.43** |
| **X → Y** | | | | | | | | |
| Cs, Uk | 14.40 | **14.59** | **14.69** | **15.28** | 22.62 | 22.60 | 22.58 | **23.82** |

*Table 6.* WMT'24 (Non Contaminated Dataset) Absolute Values with Confidence Intervals by Language Pair for 1B and 8B Models, Late Contamination, 1 Copy

| | 1B Model | | | | 8B Model | | | |
|---|---|---|---|---|---|---|---|---|
| | Baseline | Source | Target | Full | Baseline | Source | Target | Full |
| **EN → X** | | | | | | | | |
| De | 17.39 | **17.48** | **17.63** | **17.58** | 24.01 | **24.26** | **24.54** | **25.05** |
| Ru | 10.57 | **11.04** | **11.19** | **11.12** | 13.23 | **13.46** | **13.61** | **14.48** |
| Uk | 11.40 | **11.54** | **11.41** | **11.83** | 18.93 | 18.88 | **19.05** | **19.84** |
| Ja | 12.38 | **12.71** | **13.98** | **14.32** | 17.60 | 16.55 | 15.91 | 17.04 |
| Zh | 5.74 | **5.96** | **6.65** | **6.40** | 5.53 | **5.80** | **5.82** | **6.07** |
| **X → Y** | | | | | | | | |
| Cs, Uk | 14.40 | **14.88** | **14.64** | **14.81** | 22.62 | 22.27 | 22.45 | **23.39** |

*Table 7.* WMT'24 (Non Contaminated Dataset) Absolute Values with Confidence Intervals by Language Pair for 1B and 8B Models, Middle Contamination, 1 Copies

| | 1B Model | | | | 8B Model | | | |
|---|---|---|---|---|---|---|---|---|
| | Baseline | Source | Target | Full | Baseline | Source | Target | Full |
| **EN → X** | | | | | | | | |
| De | 17.39 | **17.99** | **17.80** | **18.64** | 24.01 | 23.77 | **24.14** | **25.87** |
| Ru | 10.57 | **10.93** | **11.36** | **12.11** | 13.23 | **13.84** | **14.26** | **16.64** |
| Uk | 11.40 | 11.34 | **11.79** | **12.61** | 18.93 | 18.73 | **19.74** | **21.91** |
| Ja | 12.38 | **14.46** | **13.94** | **15.19** | 17.60 | **18.99** | **19.27** | **20.36** |
| Zh | 5.74 | **6.07** | **6.55** | **7.47** | 5.53 | **5.64** | **6.09** | **7.27** |
| **X → Y** | | | | | | | | |
| Cs, Uk | 14.40 | **14.56** | **14.89** | **15.91** | 22.62 | 22.56 | **22.79** | 24.43 |

*Table 8.* WMT'24 (Non Contaminated Dataset) Absolute Values with Confidence Intervals by Language Pair for 1B and 8B Models, Late Contamination, 10 Copies

| | 1B Model | | | | 8B Model | | | |
|---|---|---|---|---|---|---|---|---|
| | Baseline | Source | Target | Full | Baseline | Source | Target | Full |
| **EN → X** | | | | | | | | |
| De | 17.39 | 17.34 | 17.18 | **17.83** | 24.01 | **24.18** | **24.40** | **25.56** |
| Ru | 10.57 | **10.68** | **10.71** | **11.57** | 13.23 | **13.37** | **13.83** | **14.99** |
| Uk | 11.40 | **11.51** | 11.38 | **11.76** | 18.93 | 18.86 | **18.99** | **19.90** |
| Ja | 12.38 | **13.58** | **14.59** | **16.53** | 17.60 | 16.81 | **18.22** | **18.42** |
| Zh | 5.74 | **5.95** | **6.03** | **6.33** | 5.53 | 5.37 | **5.71** | **6.00** |
| **X → Y** | | | | | | | | |
| Cs, Uk | 14.40 | 14.37 | **14.56** | **14.96** | 22.62 | 22.52 | **22.86** | 24.21 |

*Table 9.* WMT'24 (Non Contaminated Dataset) Absolute Values with Confidence Intervals by Language Pair for 1B and 8B Models, Middle Contamination, 10 Copies

| | 1B Model | | | | 8B Model | | | |
|---|---|---|---|---|---|---|---|---|
| | Baseline | Source | Target | Full | Baseline | Source | Target | Full |
| **EN → X** | | | | | | | | |
| De | 17.39 | **17.90** | **18.12** | **19.19** | 24.01 | 23.82 | 23.80 | **26.14** |
| Ru | 10.57 | **11.08** | **11.31** | **12.24** | 13.23 | **13.97** | **14.00** | **16.88** |
| Uk | 11.40 | **11.64** | **11.83** | **12.91** | 18.93 | 18.90 | **19.93** | **21.86** |
| Ja | 12.38 | **12.71** | **13.95** | **14.64** | 17.60 | **17.86** | **20.47** | **21.42** |
| Zh | 5.74 | **5.98** | **5.98** | **7.54** | 5.53 | **5.62** | **5.82** | **7.38** |
| **X → Y** | | | | | | | | |
| Cs, Uk | 14.40 | **15.14** | **14.69** | **16.62** | 22.62 | **22.67** | **22.81** | **24.80** |

*Table 10.* WMT'24 (Non Contaminated Dataset) Absolute Values with Confidence Intervals by Language Pair for 1B and 8B Models, Late Contamination, 100 Copies

| | 1B Model | | | | 8B Model | | | |
|---|---|---|---|---|---|---|---|---|
| | Baseline | Source | Target | Full | Baseline | Source | Target | Full |
| **EN → X** | | | | | | | | |
| De | 17.39 | 17.30 | 17.34 | **18.02** | 24.01 | **24.08** | 24.16 | **25.38** |
| Ru | 10.57 | **10.95** | **11.17** | **11.58** | 13.23 | **13.49** | **13.60** | **14.81** |
| Uk | 11.40 | **11.46** | 10.99 | **11.42** | 18.93 | **18.94** | **18.96** | **20.40** |
| Ja | 12.38 | **12.90** | **17.61** | **15.33** | 17.60 | 17.25 | 16.35 | **17.82** |
| Zh | 5.74 | 5.48 | **6.31** | **7.01** | 5.53 | **5.72** | 5.31 | **6.22** |
| **X → Y** | | | | | | | | |
| Cs, Uk | 14.40 | **14.55** | 14.30 | **15.37** | 22.62 | 22.46 | **22.73** | **24.06** |

*Table 11.* WMT'24 (Non Contaminated Dataset) Absolute Values with Confidence Intervals by Language Pair for 1B and 8B Models, Middle Contamination, 100 Copies

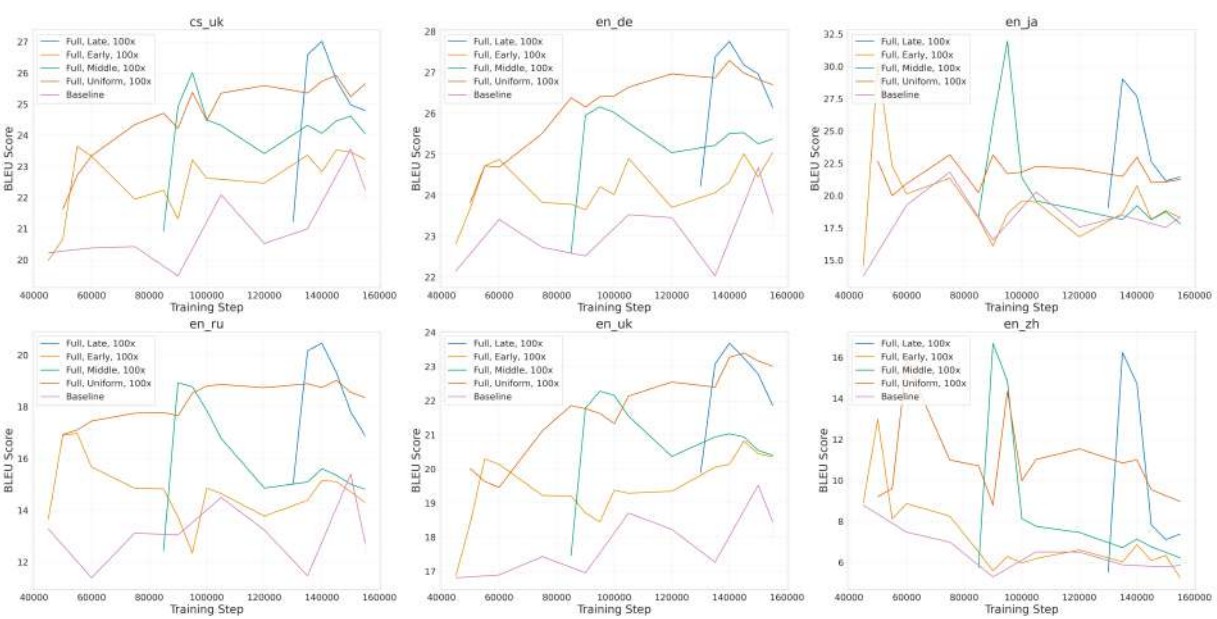

*Figure 17.* WMT'24 BLEU scores through training for 100 Copies of *Full* contamination

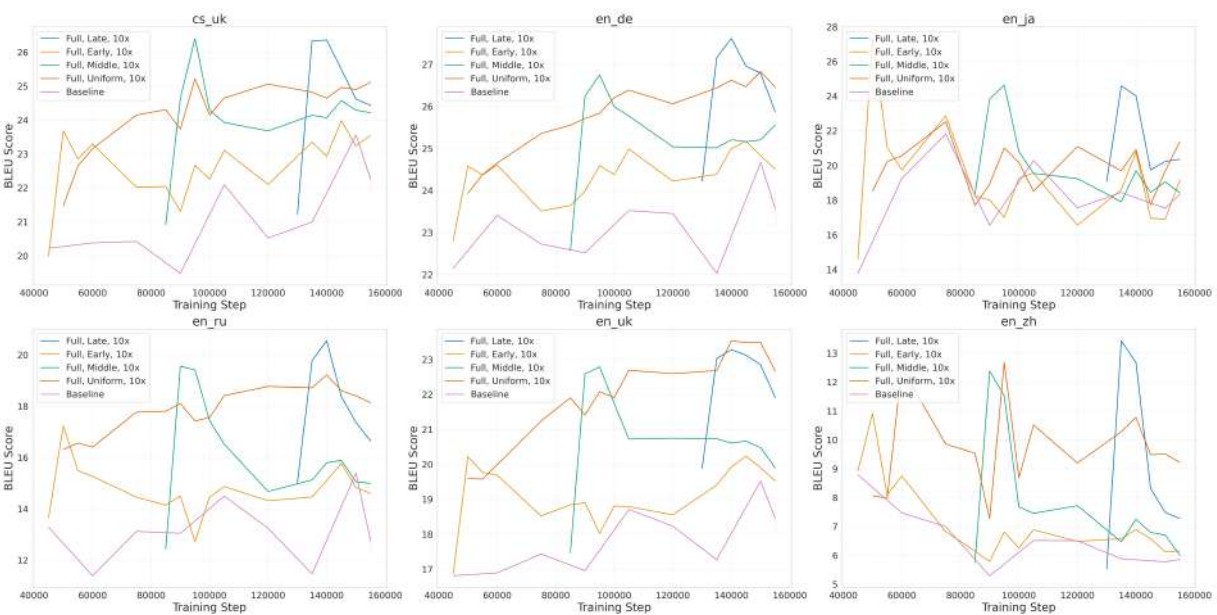

*Figure 18.* WMT'24 BLEU scores through training for 10 Copies of *Full* contamination

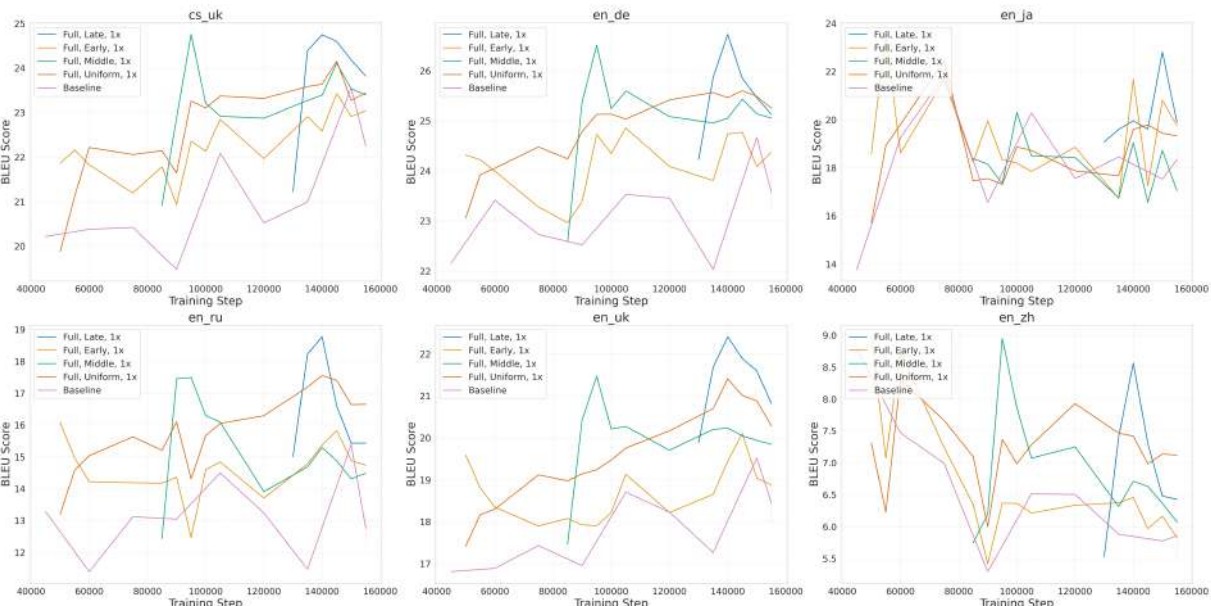

*Figure 19.* WMT'24 BLEU scores through training for 1 Copies of *Full* contamination

# F. Appendix: Impact of *Full* contamination on WMT'23 Language Pairs

| | 1B Model | | | | 8B Model | | | |
|---|---|---|---|---|---|---|---|---|
| | Baseline | Source | Target | Full | Baseline | Source | Target | Full |
| **EN → X** | | | | | | | | |
| De | 5.83 | **4.86** | **4.88** | **4.76** | 3.64 | **3.63** | 3.35 | 3.27 |
| Ru | 6.10 | **5.96** | 6.04 | **5.77** | 4.93 | 5.04 | 5.50 | **4.31** |
| Cs | 7.30 | **7.26** | 7.33 | **6.86** | 4.50 | 4.82 | 4.53 | **4.19** |
| Uk | 8.15 | 8.29 | **8.12** | 7.70 | 5.27 | **5.02** | 4.76 | 3.89 |
| He | 10.62 | **9.96** | 9.97 | 9.79 | 5.54 | **5.54** | 5.30 | 4.91 |
| Ja | 10.27 | 10.54 | 10.40 | 9.85 | 5.67 | **5.17** | 5.27 | 4.59 |
| **X → EN** | | | | | | | | |
| De | 7.34 | **7.08** | 7.11 | 6.67 | 5.21 | **5.09** | 5.11 | 4.37 |
| Ru | 9.61 | **8.76** | 8.71 | 8.40 | 6.97 | 7.13 | 6.94 | 5.59 |
| Uk | 11.11 | 11.70 | 12.14 | 11.18 | 8.11 | **7.89** | 7.71 | 7.57 |
| He | 8.95 | 9.24 | 9.28 | **8.88** | 5.26 | 5.67 | 5.45 | 5.26 |
| Zh | 7.86 | 8.17 | **7.55** | 7.50 | 5.52 | **5.34** | 5.51 | 4.73 |
| Ja | 10.36 | **9.90** | 9.87 | 9.96 | 8.45 | 8.60 | 8.74 | 7.79 |
| **X → Y** | | | | | | | | |
| Cs, Uk | 11.01 | **10.68** | 10.64 | 10.46 | 6.06 | 6.71 | 6.32 | **4.75** |

*Table 12.* Absolute MetricX (lower is better) by Language Pair for 1B and 8B Models, Late Contamination, 1 Copy

| | 1B Model | | | | 8B Model | | | |
|---|---|---|---|---|---|---|---|---|
| | Baseline | Source | Target | Full | Baseline | Source | Target | Full |
| **EN → X** | | | | | | | | |
| De | 5.83 | **5.05** | **4.90** | **4.38** | 3.64 | 3.68 | 3.37 | **2.38** |
| Ru | 6.10 | **5.98** | 6.10 | **5.23** | 4.93 | 4.99 | 5.07 | **3.31** |
| Cs | 7.30 | **7.14** | 7.16 | **6.08** | 4.50 | 4.65 | **4.44** | **3.04** |
| Uk | 8.15 | 8.18 | **7.90** | **6.63** | 5.27 | **5.08** | 4.91 | **3.10** |
| He | 10.62 | **9.59** | 9.84 | **8.95** | 5.54 | **5.30** | 5.24 | **3.98** |
| Ja | 10.27 | **10.22** | 9.85 | **8.85** | 5.67 | **5.43** | 5.12 | **3.51** |
| **X → EN** | | | | | | | | |
| De | 7.34 | **7.01** | 7.15 | **6.07** | 5.21 | **4.94** | 4.96 | **3.75** |
| Ru | 9.61 | **8.75** | 8.87 | **7.30** | 6.97 | 7.46 | 7.04 | **4.56** |
| Uk | 11.11 | 11.88 | 12.39 | **10.86** | 8.11 | 8.54 | 8.15 | **6.76** |
| He | 8.95 | **8.93** | 9.55 | **7.63** | 5.26 | 5.49 | 5.46 | **4.04** |
| Zh | 7.86 | **7.45** | 7.58 | **6.94** | 5.52 | **5.38** | 5.21 | **4.28** |
| Ja | 10.36 | **10.14** | 10.00 | **9.06** | 8.45 | 8.59 | 8.72 | **6.52** |
| **X → Y** | | | | | | | | |
| Cs, Uk | 11.01 | **10.72** | 10.93 | **9.52** | 6.06 | 6.78 | 6.38 | **3.50** |

*Table 13.* Absolute MetricX (lower is better) by Language Pair for 1B and 8B Models, Late Contamination, 10 Copies

| | 1B Model | | | | 8B Model | | | |
|---|---|---|---|---|---|---|---|---|
| | Baseline | Source | Target | Full | Baseline | Source | Target | Full |
| **EN → X** | | | | | | | | |
| De | 5.83 | **5.22** | **5.22** | **4.23** | 3.64 | **3.51** | 3.53 | **2.21** |
| Ru | 6.10 | **6.00** | 6.12 | **4.80** | 4.93 | 5.06 | **4.88** | **3.01** |
| Cs | 7.30 | **7.10** | 7.19 | **5.84** | 4.50 | 4.76 | **4.47** | **2.65** |
| Uk | 8.15 | 8.19 | 8.18 | **5.90** | 5.27 | **4.85** | 4.40 | **2.76** |
| He | 10.62 | **9.86** | 10.08 | **8.76** | 5.54 | **5.54** | 5.16 | **3.41** |
| Ja | 10.27 | 10.29 | 10.24 | **8.66** | 5.67 | **5.27** | 4.88 | **3.02** |
| **X → EN** | | | | | | | | |
| De | 7.34 | **7.11** | 7.01 | **5.60** | 5.21 | **4.98** | 4.99 | **3.56** |
| Ru | 9.61 | **8.80** | 8.85 | **7.19** | 6.97 | 7.01 | 6.80 | **4.28** |
| Uk | 11.11 | 12.25 | 11.78 | **9.96** | 8.11 | 8.22 | **7.82** | **5.00** |
| He | 8.95 | **8.86** | 9.08 | **7.42** | 5.26 | 5.56 | 5.46 | **3.12** |
| Zh | 7.86 | **7.50** | 7.84 | **6.54** | 5.52 | **5.26** | 5.89 | **4.28** |
| Ja | 10.36 | **9.84** | 10.02 | **8.88** | 8.45 | 8.44 | 8.36 | **6.32** |
| **X → Y** | | | | | | | | |
| Cs, Uk | 11.01 | 11.10 | **10.68** | **9.36** | 6.06 | 6.75 | **5.84** | **3.04** |

*Table 14.* Absolute MetricX (lower is better) by Language Pair for 1B and 8B Models, Late Contamination, 100 Copies

| | 1B Model | | | | 8B Model | | | |
|---|---|---|---|---|---|---|---|---|
| | Baseline | Source | Target | Full | Baseline | Source | Target | Full |
| **EN → X** | | | | | | | | |
| De | 21.71 | 21.59 | 21.52 | **23.37** | 30.95 | **31.95** | **32.11** | **34.34** |
| Ru | 13.42 | **13.89** | 13.88 | **14.78** | 15.42 | 14.53 | 14.77 | **16.97** |
| Cs | 14.06 | **14.65** | 14.57 | **15.67** | 22.65 | 23.75 | 24.28 | **25.65** |
| Uk | 12.94 | 12.60 | 12.83 | **13.59** | 21.96 | 21.16 | 21.42 | **24.31** |
| He | 9.72 | **10.00** | 10.02 | **10.69** | 18.27 | **18.76** | 18.51 | **20.02** |
| Ja | 4.47 | **5.04** | 5.51 | **5.23** | 4.97 | **5.11** | 4.90 | 4.79 |
| **X → EN** | | | | | | | | |
| De | 26.46 | 26.21 | 26.02 | **27.32** | 33.59 | **34.03** | 33.87 | **37.15** |
| Ru | 22.55 | **22.96** | 22.99 | **23.87** | 28.41 | 27.60 | 27.92 | **30.50** |
| Uk | 20.23 | 20.17 | 19.31 | **20.33** | 27.55 | **28.23** | 26.65 | **28.20** |
| He | 25.05 | **25.14** | 24.41 | **26.68** | 38.62 | 37.53 | 37.88 | **40.99** |
| Zh | 11.31 | **11.34** | 11.57 | **12.70** | 19.81 | 18.93 | 19.31 | **21.44** |
| Ja | 4.97 | **5.22** | 5.53 | **5.83** | 10.42 | 9.89 | 10.32 | **12.40** |
| **X → Y** | | | | | | | | |
| Cs, Uk | 11.24 | **11.68** | 11.32 | **12.03** | 20.39 | 19.44 | 19.51 | **22.67** |

*Table 15.* Absolute BLEU by Language Pair for 1B and 8B Models, Late Contamination, 1 Copies

| | 1B Model | | | | 8B Model | | | |
|---|---|---|---|---|---|---|---|---|
| | Baseline | Source | Target | Full | Baseline | Source | Target | Full |
| **EN → X** | | | | | | | | |
| De | 21.71 | **21.82** | 21.30 | **25.60** | 30.95 | **31.57** | 33.09 | **40.64** |
| Ru | 13.42 | **13.84** | 13.76 | **16.93** | 15.42 | 14.25 | 15.08 | **22.56** |
| Cs | 14.06 | **14.46** | 14.80 | **17.57** | 22.65 | 23.18 | 24.82 | **33.93** |
| Uk | 12.94 | 12.28 | 13.44 | **16.19** | 21.96 | 21.84 | 23.40 | **31.58** |
| He | 9.72 | **9.94** | 10.36 | **12.33** | 18.27 | **18.42** | 19.42 | **25.43** |
| Ja | 4.47 | **5.14** | 6.13 | 6.46 | 4.97 | **5.66** | 4.57 | **8.39** |
| **X → EN** | | | | | | | | |
| De | 26.46 | 26.11 | 26.43 | **29.96** | 33.59 | **33.69** | 34.71 | **42.60** |
| Ru | 22.55 | **23.03** | 22.99 | **26.67** | 28.41 | 27.41 | 28.07 | **34.38** |
| Uk | 20.23 | 19.93 | 19.44 | **22.37** | 27.55 | 27.53 | 26.65 | **34.72** |
| He | 25.05 | **25.11** | 25.06 | **31.50** | 38.62 | 37.95 | 39.00 | **49.90** |
| Zh | 11.31 | 11.05 | **11.64** | **14.38** | 19.81 | 19.41 | 19.62 | **27.25** |
| Ja | 4.97 | **5.61** | 5.90 | 7.27 | 10.42 | 10.29 | 10.76 | **17.11** |
| **X → Y** | | | | | | | | |
| Cs, Uk | 11.24 | 11.15 | **11.82** | **14.89** | 20.39 | 19.29 | 20.12 | **29.12** |

*Table 16.* Absolute BLEU by Language Pair for 1B and 8B Models, Late Contamination, 10 Copies

| | 1B Model | | | | 8B Model | | | |
|---|---|---|---|---|---|---|---|---|
| | Baseline | Source | Target | Full | Baseline | Source | Target | Full |
| **EN → X** | | | | | | | | |
| De | 21.71 | 21.24 | 21.20 | **28.70** | 30.95 | **31.09** | 33.13 | **47.55** |
| Ru | 13.42 | **13.74** | 14.80 | **19.15** | 15.42 | 14.54 | **15.53** | **27.61** |
| Cs | 14.06 | **14.73** | 15.57 | **19.93** | 22.65 | 22.86 | 26.02 | **44.86** |
| Uk | 12.94 | 12.24 | 13.70 | **18.83** | 21.96 | 22.18 | 24.02 | **40.30** |
| He | 9.72 | **10.05** | 10.27 | **13.61** | 18.27 | 18.11 | 20.23 | **33.56** |
| Ja | 4.47 | **4.67** | 5.92 | **5.98** | 4.97 | **6.71** | 4.48 | **9.87** |
| **X → EN** | | | | | | | | |
| De | 26.46 | 26.36 | **27.11** | **32.06** | 33.59 | 33.47 | **35.66** | **47.56** |
| Ru | 22.55 | **23.13** | 23.49 | **28.08** | 28.41 | 27.63 | 28.66 | **38.08** |
| Uk | 20.23 | **20.59** | 20.20 | **23.11** | 27.55 | **27.57** | 27.68 | **40.71** |
| He | 25.05 | 24.65 | 24.74 | **34.11** | 38.62 | 36.72 | 39.94 | **59.93** |
| Zh | 11.31 | **11.39** | 11.37 | **15.96** | 19.81 | 19.16 | 19.62 | **32.56** |
| Ja | 4.97 | **5.15** | 5.77 | **8.82** | 10.42 | 10.23 | **11.34** | **22.94** |
| **X → Y** | | | | | | | | |
| Cs, Uk | 11.24 | **11.35** | 11.75 | **17.63** | 20.39 | 18.88 | **21.23** | **37.34** |

*Table 17.* Absolute BLEU by Language Pair for 1B and 8B Models, Late Contamination, 100 Copies

# G. Appendix: Search and Decontamination

For the search and decontamination step we search the tokenized pre-training corpus with the bpe tokenized examples. We first split the evaluation examples into 8-grams and then

search each 8-gram. Once we find all matches for all 8-grams in the pre-training corpora we extend each match to the left and right as much as we can. Then we take the longer token match similar to Singh et al. (2024). Once we have the longest match we calculate the number of overlapping tokens. We use the overlapping percentage of tokens in both the source and target field to filter examples as contaminated or not.

Decontaminating these examples could be done by either removing the relevant examples from the training data or removing these examples from the test set. After analyzing the overlap statistics we choose to do the latter and decontaminate our test set by removing these examples from the test set.

First we present histograms of contamination scores for all language pairs in WMT'23 and the three language pairs from FLORES that we use in our testing. Here we observe that language pairs from FLORES have more contamination compared to language pairs from WMT'23. Second we observe that contamination in WMT'23 seems to be mostly partial. While the relationship with partial contamination and performance is still studied, for practical reasons we followed an pre-established cutoff of 0.7 that was introduced by Chowdhery et al. (2022).

When we check for the threshold in any field source or target and analyze the test sets we see in Figure 20 that the actual examples that are contaminated are around 10% of the test data where most of these examples are what we can call single field contamination(contamination in just source or just target) There is a small subset of examples where we see both the source and the target is contaminated. When we analyzed this subset we observed that they are generally simple sentence structures that can frequently occur on the internet. In Table 18 we are presenting some examples that are in this group.

We also check the WMT'24 test data and the contamination level with our main corpus and find that 162 examples are contaminated in their source field above the threshold 0.7 and we remove these examples from the WMT'24 test data that we use for our experiments.

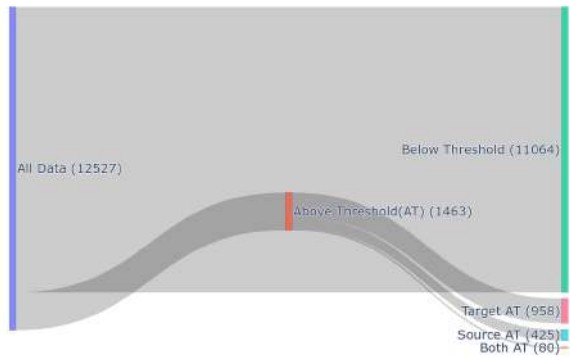

*Figure 20.* Riverplot of clean, contaminated and different forms of contamination for the threshold 0.7.

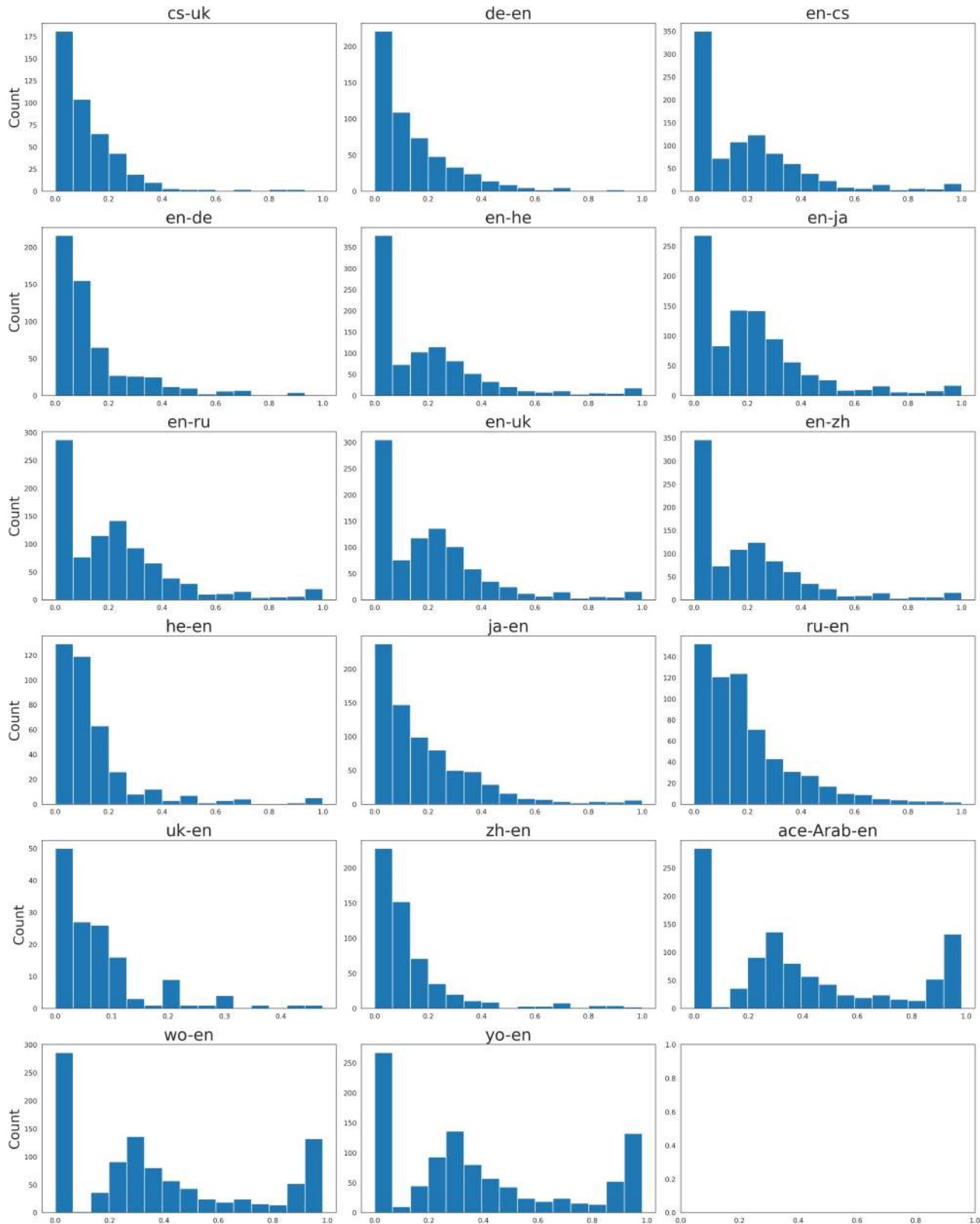

*Figure 21.* Histogram of contamination scores. Here the scores are calculates as $max(s_{source}, s_{target})$ where $s_{source}$ is the percentage of overlapping tokens with the longest contamination that contains any tokens from the source field and $s_{target}$ similarly for target.

| Indx | Source | Target |
|------|--------|--------|
| 1 | schwarze, lange Haare mit Pferdeschwanz | long, black hair with a ponytail |
| 2 | Sieben Leichtverletzte nach Auffahrunfall auf A5 | Seven slightly injured in rear-end collision on A5 |
| 3 | Schlossplatz Stuttgarter gedenken der Opfer der Erdbebenkatastrophe | Schlossplatz Stuttgart commemorate the victims of the earthquake disaster |
| 4 | 3D Drucker Prusa i3 DIY | 3D Printer Prusa i3 DIY |
| 5 | Photos from my visit to Ghana in 2011. | Fotografie z mé návštěvy Ghany v roce 2011. |
| 6 | Was it the 2012 Apple Maps disaster? | Byla to katastrofa pro Apple Maps v roce 2012? |
| 7 | OLIVIER DOULIERY/AFP via Getty Images | OLIVIER DOULIERY / AFP prostřednictvím služby Getty Images |
| 8 | For more information see our Privacy Policy. | Další informace naleznete v našich Zásadách ochrany soukromí. |
| 9 | First medical aid is also provided on a round-the-clock basis. | První lékařská pomoc je poskytována nepřetržitě. |
| 10 | Follow Metro Sport for the latest news on Facebook, Twitter and Instagram. | Sledujte Metro Sport a jeho nejnovější zprávy na Facebooku, Twitteru a Instagramu. |
| 11 | This is the best way to earn the press's trust. | Ten nejlepší způsob, jak si získat důvěru tisku. |
| 12 | Follow him on Instagram: @awr_hawkins. | Sledujte ho na Instagramu: @awr_hawkins. |
| 13 | Reach him directly at awrhawkins@breitbart.com. | Obraťte se přímo na něj na adrese awrhawkins@breitbart.com. |
| 14 | It handed him a nine-month suspended sentence and a fine. | Soud mu uložil devítiměsíční podmíněný trest a pokutu. |
| 15 | I give this book 10 stars! | Této knize dávám 10 hvězdiček! |
| 16 | Good, but would like to find something better | Dobré, ale chtělo by to najít něco lepšího |
| 17 | Good umbrella, would buy it again if I had to | Dobrý deštník, koupil bych si ho znovu, kdybych musel |
| 18 | but it happens enough to be annoying. | ale stává se to docela často na to, aby to bylo nepříjemné. |
| 19 | Nothing like the previous Stylo phones, MASSIVE DISAPPOINTMENT. | Nic jako předchozí telefony Stylo, VELKÉ ZKLAMÁNÍ. |
| 20 | Some Super Bowl Commercials I Can't Wait to See | Einige Super Bowl-Werbespots, die ich unbedingt sehen will |

*Table 18.* Examples where both source and target are contaminated above the 0.7 threshold.

