# OpenReview forum: "Overestimation in LLM Evaluation: A Controlled Large-Scale Study on Data Contamination’s Impact on Machine Translation"
_ICML.cc/2025/Conference — ICML 2025 poster_

### Official Review · Reviewer_QNfP · 2025-02-17

**Overall Recommendation:** 4

**Summary:**

This paper presents a controlled study on the impact of data contamination on machine translation. They decontaminate their train-test splits and train two decoder-only models of different sizes (with 1 and 8 billion parameters). Then, they add test data into the pretraining data and train a contaminated model branching out from the baseline checkpoint. Finally, they compare the relative performance of the contaminated and the baseline model on contaminated and non-contaminated data. Their main findings are as follows:

(i) contaminating with both source and target sides leads to substantial performance inflation on those test sets;

(ii) partial contamination (source-only or target-only) leads to smaller/less consistent inflation;

(iii) the temporal persistence of contamination matters;

(iv) contamination has more impact on larger models;

(v) contamination requires sufficient language representation to have a measurable effect.

**Claims And Evidence:**

Their main claims/findings are summarized above (i-v); they are all well-supported through experiments. (iv) seems to hold in practice but this is only tested for two model sizes (this is acknowledged in the paper).

**Essential References Not Discussed:**

NA.

**Experimental Designs Or Analyses:**

The experimental design is sound. In particular the authors searched and found that ~10% of test set examples were already contaminated (Section 3.1). This shows the importance of their decontamination step (the first step in Fig. 1), which is missing in previous work according to Table 1.

**Methods And Evaluation Criteria:**

The method seems appropriate. Using branching to save computation sounds reasonable. It would be nice to explore other types of contamination (eg., paraphases) but this is not really needed. However, I can’t understand the reason for drawing conclusions based on BLEU scores. Its issues are well known, check for instance [1, 2]. The authors include Metric-X results in the Appendix, so my question is:

(1) Is there a reason to report BLEU in the main paper and Metric-X results only in the Appendix?

[1] Results of WMT22 Metrics Shared Task: Stop Using BLEU – Neural Metrics Are Better and More Robust (Freitag et al., WMT 2022)

[2] Results of WMT23 Metrics Shared Task: Metrics Might Be Guilty but References Are Not Innocent (Freitag et al., WMT 2023)

**Other Comments Or Suggestions:**

Minor comments:

- Typo in L208 (missing white space after “language pairs:”)
- It would be valuable to expand a bit the discussion on the broader implications of these findings, eg., how to adapt existing evaluation protocols/benchmarks to mitigate this issue?

**Other Strengths And Weaknesses:**

In addition to what I pointed out above, the paper is exceptionally well-written and is easy to follow; the analysis setup is sound, and most of their choices are well-justified. The main weakness is the reliance on BLEU as the main evaluation metric to draw conclusions. I’d like to see some discussion justifying this choice.

**Questions For Authors:**

See above.

**Relation To Broader Scientific Literature:**

Table 1 provides a good summary, but I think this is the first work doing this analysis for machine translation.

**Theoretical Claims:**

No proofs are presentend but I think none are needed (the paper focuses on empirical findings).

---

> ### Author Rebuttal · Authors · 2025-04-01
>
> We thank the reviewer for their positive review.  We will extend the discussion section to include a discussion on the broader implications of our findings.
>
> On the choice of BLEU as a metric: We acknowledge that a string-based metric (like BLEU) has limitations. For this reason, we already accompany all of our evaluations with an additional learned metric (i.e., MetricX) which has been widely adopted by the field. All drawn conclusions are consistent across both reported metrics. Our choice to surface BLEU in the main paper is based on its wider adoption within the ML community.

---

### Official Review · Reviewer_x73P · 2025-03-10

**Overall Recommendation:** 3

**Summary:**

This paper investigates the impact of data contamination on machine translation. In particular, the paper tests factors including source contamination, target contamination and temporal distribution.

**Claims And Evidence:**

The claims are supported by abundant experiments from multiple data source.

**Essential References Not Discussed:**

No missing reference found.

**Experimental Designs Or Analyses:**

No significant problems in experiments or analyses.

**Methods And Evaluation Criteria:**

No significant flaws in method and evaluation.

**Other Comments Or Suggestions:**

See the previous comment.

**Other Strengths And Weaknesses:**

Strength: Abundant experiment to support the claim

Weakness: Although it conducts a thorough investigation on the impact of data contamination in machine translation, there is no suggestion on how to resolve this issue. Significance can be further improved by e.g., methods to detect data contamination in the pretraining data, or method to detect data contaminated pretrained models.

**Questions For Authors:**

1. Is there an efficient way to detect data contamination in the pretrained dataset, i.e., overlap between pretrained and test data?
2. Is it possible to determine whether there is data contamination, if you only have access to the pretrained model and test dataset, but no access to the pretrained data? In other words, is there a clear separation between the performance of a model pretrained with non-contaminated data, and that of a model pretrained with contaminated data?

**Relation To Broader Scientific Literature:**

This paper extends the study of data contamination to the field of machine learning.

**Theoretical Claims:**

Not applicable for this paper.

---

> ### Author Rebuttal · Authors · 2025-04-01
>
> Thanks a lot for your review of our paper.
>
> First, we respectfully disagree that the contribution of our paper is insignificant. The key assumption of data contamination is motivated by the hypothesis that consuming test data leads to an overestimation of a model’s performance. While intuitive, this hypothesis has not been rigorously tested at this scale, and our work aims to fill that gap in the literature. Further, we suspect many researchers care about testing this hypothesis but are unable to run the experiments that we did due to the computational cost. Therefore, we feel our contributions are significant and our findings are very relevant to the research community.
>
> 1. There are methods for detecting contamination by looking at the overlap between the training and test sets, including https://arxiv.org/abs/2411.03923. Our train-test decontamination stage is inspired by the findings of this paper. Our implementation is very efficient because it’s highly parallelizable and uses a Bloom filter combined with an exact search to search more than 100k 8-grams from the test data in trillions of training tokens in less than a day.
>
> 2. There are papers focused on finding contamination with just access to model activations. However this task is shown to be harder than initially expected (https://arxiv.org/abs/2402.07841.) Our experiments and findings regarding contamination in section 5.4 can partially highlight why this task can be harder than initially hypothesized. We show that models don’t naively memorize text in the pretraining data suggesting that model behaviour might not be as easily separable between what is in the pre-training data and what is not. Recent work also shows that models can verbatim generate text that is explicitly not in their pre-training data https://arxiv.org/abs/2503.17514. This suggests that models learn more general representations of language and understanding contamination and the impact on evals requires understanding knowledge formation and access in large language models.

---

> > ### Comment · Reviewer_x73P · 2025-04-02
> >
> > Thanks for you response. I agree that the experiment results you have should contribute to research in machine learning translation. I've raised the score to 3.

---

### Official Review · Reviewer_J6EW · 2025-03-13

**Overall Recommendation:** 4

**Summary:**

This paper presents a controlled large-scale study on how data contamination impacts machine translation evaluation in large language models. The researchers created a carefully decontaminated train-test split, then systematically reintroduced contamination under controlled conditions across different modes, temporal distributions, and frequencies on both 1B and 8B parameter models.

The key findings of the paper shows that the full contamination mode (where both the source and target text of the test set exist) can dramatically inflate performance metrics (up to 30 BLEU points for 8B models). Corroborating previous study, contamination effect is more evident when it is introduced later during training, but the paper also found that uniform contamination leads to the most significant effect. Last but not least, contamination requires a sufficient language representation to have a measurable effect. These findings highlight the critical importance of properly decontaminating evaluation benchmarks to avoid substantially overestimating model capabilities.

**Claims And Evidence:**

Following the breakdown from the introduction of the paper:

- Contaminating source-target MT pairs inflates performance on those test sets. -> adequately supported
- The temporal distribution of contamination matters. -> adequately supported
- The impact of contamination increases with model scale. -> supported with limitation (only data points for 1B & 8B is supplied), but acknowledged in the paper
- Contamination requires sufficient language representation to have a measurable effect -> adequately supported

**Essential References Not Discussed:**

The paper has already cited [1]. I would recommend also discussing [2] because they are tackling a similar data contamination problem from a different (the system user's) angle, even though it predates the boom of LLMs. [3] and [4] should also be discussed in the context of alternative decontamination methods, which is something I would like to see added (more details in "Other Comments Or Suggestions").

**Experimental Designs Or Analyses:**

I checked everything that's presented in Section 3 and 4 carefully. While I don't spot issues with what's presented, I find the level of details given for checkpoint-averaging a little lacking. See more details in "Other Strengths And Weaknesses".

**Methods And Evaluation Criteria:**

Yes

**Other Comments Or Suggestions:**

I would like to see two main things added in the next draft.

### More details on checkpoint branching

I would like to see Section 3.4 significantly re-written to better present this method.

* Start with the motivation. Stress what you have given up compared to multiple full re-training runs (essentially, you have fixed initialization seed and data sampling order), and what you get out of this (re-use some compute from other runs).
* When you introduce a "branch", how do you implement it? Do you just add extra samples to the training data? What about learning rate scheduler?
* How do you introduce uniform contamination?
* In total, how many "branches" did you have to introduce for one set of experiment? (e.g. for 1B, single-copy, full contamination)

### More discussions on the choice of decontamination method

In its current shape, the paper only mentions their choice of decontamination method in Appendix G. I think this is an important experiment design that is worth talking in the main paper as well. Moreover, this should be treated as an intentional choice, since what this paper adopts is not the only contamination detection/decontamination method out there (see [1][2][3][4] above). The paper should acknowledge these other methods and discuss how the other methods relate to their method.

I also spot a few minor typos/style comments:

* L74 left col. -- "up to 60 BLEU points", not 30?
* L83 right col. -- "previous works" -> previous work
* L85-86 right col. -- use -> used
* L205-216 left col. -- for readers that's not familiar with how WMT datasets are set up, probably give some justification for why you take 2023 as contaminated dataset and 2024 as non-contaminated dataset? (But then you also report in Appendix G that there are also some contamination in 2024 dataset, which I'm not sure how it happened?)
* Figure 2 -- I'm confused about how the methods are ranked? Specifically, there seems to be some methods below "baseline" that's doing worse than baseline? (Same with Figure 3)

**Other Strengths And Weaknesses:**

This is a high-quality controlled study on the effect of data contamination on the inflation of machine translation evaluation result. The problem studied here has significant real-world implications and is of value to other tasks other than machine translation as well. The presentation quality of the paper is high, with the core methods and findings very clearly presented.

The main complaints I have about the paper is the lack of details in the checkpoint branching method (Section 3.4). This is the most significant experimental design choice of this paper, and yet is very briefly glossed over. This is not at all sufficient for subsequent studies to reproduce their results.

**Questions For Authors:**

Please clarify the following:

* The questions I listed in "Other Comments Or Suggestions" about checkpoint branching
* The ranking of setups in Figure 2
* What does "threshold of 0.7" mean in Appendix G (L1343 and L1359) -- basically, here is the relevant quote I found from the section 8 of the PALM paper that was cited:

> "So, we were able to split each dataset into a “contaminated” and “clean” subset based on whether at least 70% of the 8-grams in question, prompt, or target were seen at least once our training data."

but I was confused about how exactly it is computed, and how it related to what's been discussed there.

**Relation To Broader Scientific Literature:**

There are existing work that looks at membership detection of certain data point into a models pre-training data [1][2][3][4], each of which has established some sort of detection algorithm: min k% prob [1], membership inference attacks [2], and bloom filters [3], infini-gram [4]. There are also papers that examine the impact of data contamination on fair LLM evaluation [5][6]. However, to the best of my knowledge, none of them has studied systematically how introducing data contamination in the pre-training data in different ways can introduce different effects in the resulting model, so I think the novelty of the findings in this paper is significant.

- [1] https://arxiv.org/pdf/2310.16789
- [2] https://aclanthology.org/2020.tacl-1.4.pdf
- [3] https://arxiv.org/pdf/2303.03919
- [4] https://arxiv.org/pdf/2401.17377
- [5] https://aclanthology.org/2023.findings-emnlp.722.pdf
- [6] https://aclanthology.org/2024.findings-acl.716.pdf

**Theoretical Claims:**

N/A

---

> ### Author Rebuttal · Authors · 2025-04-01
>
> We thank the reviewer for the detailed review and insightful comments. We will use the reviewer’s feedback to revise the writing of Section 3.4 and include missing citations.
>
> 1. Details on contamination score calculation: The contamination score is calculated as the number of tokens in the longest overlap, over to the number of tokens in the eval example. If more than 70% of the tokens are contained in the longest matching overlap we label an example as contaminated and remove it from our test set.
> 2. Choice of WMT23/24 datasets: The WMT23/24 datasets used in our experiments are the standardized datasets, established by WMT (Conference on Machine Translation) annual conference.
> 3. Contamination from WMT’24: Source segments from WMT’24 are generally standard pieces of texts that are frequently found in news articles or general webtext. Since WMT sources input segments from online sources, it is possible that frequently used sentences might be found in any pre-training data. For this reason, we make sure that we run decontamination tests for all datasets we use.
> 4. On the ranking of Figure 2: The presented methods are sorted based on the average performance across language pairs. They are shown in descending order for both Figs 2 and 3. We will clarify this in the next version of the paper.
> 5. Checkpoint branching details: In our branching method, each branch is effectively a copy of the baseline model, with all the optimizer and meta data inherited from the checkpoint. As training continues, the only difference is the pre-training mixture that the new model (continued pre-training) is exposed to. This mixture is a copy of the baseline one with random examples replaced by the contaminated instances. Our setup covers 42 contamination experiments, leading us to 42 “branches”.  We will include those details in the next version.

---

> > ### Comment · Reviewer_J6EW · 2025-04-03
> >
> > Thanks for the response. Minor clarification about my request:
> >
> > > probably give some justification for why you take 2023 as contaminated dataset and 2024 as non-contaminated dataset?
> >
> > What I'm trying to suggest is that you could point out WMT shared task organizers do collect data with recent time stamp to avoid the test set appearing in previous pre-trained data, which is why this is a good setup for data contamination study.
> >
> > > it is possible that frequently used sentences might be found in any pre-training data.
> >
> > Good point. Thanks for clarifying.

---

### Official Review · Reviewer_7tAa · 2025-03-14

**Overall Recommendation:** 3

**Summary:**

The paper analyzes the influence of data contamination on LLMs trained for machine translation.  Their testing controls factors such as the modes of contamination, the temporal distribution of contaminated samples, and the frequency the contaminated samples are presented.  From their experimentation they demonstrate that (1) parallel data contamination provides a much larger impact than monolingual data contamination, (2) uniformly distributed data contamination has the most persistent impact, (3) larger models are more influenced by data contamination, and (4) data contamination is most prevalent once the model already has a language understanding capability.

**Claims And Evidence:**

Yes, claims are supported by evidence, but they would be more convincing if a wider range of models were tested on.  (Not a big deal as I understand this would require an incredible amount of compute and would most likely lead to the same outcomes.)

**Essential References Not Discussed:**

No.

**Experimental Designs Or Analyses:**

Yes, experimental design and analysis is sound.

**Methods And Evaluation Criteria:**

Yes, the methods and evaluation make sense for the problem.

**Other Comments Or Suggestions:**

None.

**Other Strengths And Weaknesses:**

Strengths:
1) Propose a checkpoint-branching scheme that reduces variance and increases computational cost of their experimentation.
2) This is the first paper that provides experimentation with data control that explores the influence of data contamination for LLMs when they are applied to machine translation.

Weaknesses:
1) The experimentation required an extensive amount of computation and there is a lack of documentation outlining the amount of compute required in case future people want to replicate the methodology for alternative seq2seq tasks.
2) A lot of the conclusions drawn from experimentation were not incredibly insightful.  For instance, (1) and (3) are already known by the community.
3) It does not seem that testing the influence of the number of copies of the test set in the training data is entirely necessary.  It seems unlikely that the training data would contain 10-100 copies of the test set ever.  Unless I am misinterpreting something these results appear as though they were included to exaggerate the influence of data contamination on machine translation.  Is this something that would be seen in real world data?
4) The conclusions were made on a single architecture, and it is unclear if the conclusions would extend to alternative architectures.
5) The testing is only done for machine translation, and it does not explore if similar results would hold for alternative seq2seq problems.

**Questions For Authors:**

1) What is the overall amount of compute required for all experiments in terms of GPU hours? How many models were trained from your checkpoint branch?
2) Do you think similar trends would carry over to alternative architectures?
3) Are there any circumstances where there might be 10-100 copies of the test set in the training set?

**Relation To Broader Scientific Literature:**

The paper contributes a thorough analysis of the influence of data contamination for machine translation LLMs.  This is a prevalent issue that has not been explored to the same level in a prior publication.

**Theoretical Claims:**

N/A

---

> ### Author Rebuttal · Authors · 2025-04-01
>
> We thank the reviewer for the detailed comments and feedback. We respond to the reviewer’s comments below.
>
> 1. Question 1 (training compute): All models are implemented as continued pre-training starting off baseline checkpoints. For instance, models with late contamination are only trained for 10% of the entire training (given we kickstart the model from the 90% of the baseline’s mark). In practice, this means that our checkpoint branching approach reduces the compute requirements about 60% (compared to the compute budget we would have spent, if all models were trained from scratch.
> The experiments being expensive is definitely a challenge however we have undertaken this cost for the community and shared our findings openly that can answer critical questions in this area as well as inform future research. We think it is an important contribution of this paper that we have committed to rigorously experimenting the impact of contamination given the computational requirements and not a weakness.
> 2. Question 2 (generalization to different architectures): We acknowledge that our findings are limited to decoder-only LLMs, which constitute the predominant LLM paradigm in recent years.
> 3. Question 3 (frequency of contamination in pre-training):  While in principle, pre-training data deduplication is a common practice; in multilingual settings, upsampling under-represented languages is an equally common tactic. The latter is dictated by the lack of resources for many languages, and inevitably can lead to data being repeated more than once. As a result, it is reasonable to assume that once a test instance bleeds into pre-training data, it might end up seen by the model more than once.
> 4. On the novelty of our findings: The more intuitive result, i.e., contamination leads to performance overestimation, has been hypothesized by many works; however, which data conditions contribute to this overestimation and to what extent, has not been studied by prior work. Our study contributes empirical evidence that contamination could, but not always lead to performance overestimation, a behavior that depends on various factors ranging from the way contaminated test sets are presented in the pre-training data, to how well the contaminated languages is represented in the pre-training distribution (Section 5 and 5.1).
> 5. On the choice of Machine Translation (MT) test sets: MT test sets naturally let us study how data contamination interacts with pre-training resource requirements (which are different across the many languages we studied with this work). As evidenced from our findings, whether contamination has a measurable impact on downstream performance does connect to the language’s representation during pre-training.

---

### Decision · Program_Chairs · 2025-05-01

**Decision:**

Accept (poster)

**Comment:**

This paper studies the problem of data contamination in LLM-based MT on two model sizes (1B and 8B scales) through controlled experiments. The results suggest that contamination with both source and target substantially inflates BLEU scores, while source-only and target-only contamination generally produce smaller, less consistent over-estimations.

While the results are mostly predictable and the novelty is fairly limited, the reviewers agree that this is a solid empirical paper whose findings are valuable to the research community to better understand the phenomenon of data contamination. I therefore recommend acceptance.